# Boosting lithium ion conductivity of antiperovskite solid electrolyte by potassium ions substitution for cation clusters

Lei Gao[1], Xinyu Zhang[2], Jinlong Zhu [2], Songbai Han [2] ✉, Hao Zhang[1], Liping Wang [2], Ruo Zhao[2], Song Gao[1], Shuai Li [2], Yonggang Wang[1], Dubin Huang[1], Yusheng Zhao[2] & Ruqiang Zou [1] ✉

Solid-state electrolytes with high ionic conductivities are crucial for the development of all-solid-state lithium batteries, and there is a strong correlation between the ionic conductivities and underlying lattice structures of solid-state electrolytes. Here, we report a lattice manipulation method of replacing $[Li_2OH]^+$ clusters with potassium ions in antiperovskite solid-state electrolyte $(Li_2OH)_{0.99}K_{0.01}Cl$, which leads to a remarkable increase in ionic conductivity ($4.5 \times 10^{-3}$ mS cm$^{-1}$, 25 °C). Mechanistic analysis indicates that the lattice manipulation method leads to the stabilization of the cubic phase and lattice contraction for the antiperovskite, and causes significant changes in Li-ion transport trajectories and migration barriers. Also, the Li||LiFePO$_4$ all-solid-state battery (excess Li and loading of 1.78 mg cm$^{-2}$ for LiFePO$_4$) employing $(Li_2OH)_{0.99}K_{0.01}Cl$ electrolyte delivers a specific capacity of 116.4 mAh g$^{-1}$ at the 150th cycle with a capacity retention of 96.1% at 80 mA g$^{-1}$ and 120 °C, which indicates potential application prospects of antiperovskite electrolyte in all-solid-state lithium batteries.

All-solid-state lithium batteries (ASSLBs), using solid-state electrolytes (SSEs) to replace organic polymer separators and liquid electrolytes in conventional Li-ion batteries, have triggered intensive research studies because of their merits of potentially improved safety[1–3]. Besides, ASSLBs based on Li-metal anodes are expected to break through the energy density bottleneck, which benefits from the compatibility of SSEs with Li-metal and the mechanical strength of SSEs to prevent the penetration of Li dendrites[4–6]. Consequently, a variety of SSEs have been investigated in recent years, including argyrodite[7,8], thio-phosphate[9,10], sodium superionic conductor (NASICON)[11,12], garnet[13,14], antiperovskite[15–17], and others[18].

Still, there are many limitations to overcome for these candidate SSEs. For example, garnet SSEs need high-temperature sintering for densification with high-energy consumption and have huge interfacial resistance against Li-metal anodes[19]. The NASICON-type SSEs would be deteriorated in contact with Li-metal due to the reduction of Ge$^{4+}$ or Ti$^{4+}$[20–22]. Although sulfide-based SSEs (e.g., Li$_{10}$GeP$_2$S$_{12}$, Li$_7$P$_3$S$_{11}$) with excellent ionic conductivity (>1 mS cm$^{-1}$) may support the operation of ASSLBs at room temperature (RT), the electrochemical instability and the harmful H$_2$S gas produced by exposure to air restrict their practical applications[5,23].

Recently, antiperovskite ionic conductors, as special SSEs, have been gaining increasing attention[15,24,25]. In the aspect of synthesis, antiperovskite (e.g., Li$_2$OHCl) SSEs could be easily prepared by one-step heating treatment of low-cost ingredients (equimolar LiOH and LiCl) and densified at low temperature with low energy consumption[25]. In terms of stability to Li-metal, the antiperovskite family has been proven to be stable to metallic lithium during the cycling of solid-state batteries[26,27]. Particularly, the low melting point (~300 °C) of antiperovskite is rare in the reported candidate SSEs[28], which inspires the concept of 'melt-infiltration' SSE technology and enables the potential for scalable, low-cost manufacturing of ASSLBs[29]. However, the

[1]School of Materials Science and Engineering, Peking University, 100871 Beijing, China. [2]Academy for Advanced Interdisciplinary Studies and Department of Physics, Southern University of Science and Technology, 518055 Shenzhen, China. ✉e-mail: hansb@sustech.edu.cn; rzou@pku.edu.cn

undesirable ionic conductivity of $Li_2OHX$ ($X$ = Cl, Br) hinders the further development of antiperovskite in ASSLBs, especially the ionic conductivity of orthorhombic $Li_2OHCl$ is only $10^{-4}$–$10^{-5}$ mS cm$^{-1}$ at RT (-25 °C)[26,30].

Keeping the crystal in the metastable phase is a conventional method to improve the ionic conductivity of SSEs. Typically, the ionic conductivity of $Li_7La_3ZrO_{12}$ was remarkably improved after the cubic phase was stabilized by Al/Ta doping[31,32]. The nano-porous structure stabilized the metastable phase of β-$Li_3PS_4$ and enhanced the ionic conductivity by 3 orders of magnitude[9]. Similarly, the ionic conductivity of $Li_2OHCl$ increased significantly after the phase transition from orthorhombic to cubic around 30−50 °C[26], which inspires us to attempt to improve the ionic conductivity of antiperovskite SSEs by lattice manipulation.

Herein, we report a lattice manipulation method of replacing cation clusters with potassium ions in antiperovskite, resulting in a remarkable increase of ionic conductivity of $(Li_2OH)_{0.99}K_{0.01}Cl$ to $4.5 \times 10^{-3}$ mS cm$^{-1}$ at 25 °C, which is 32 times higher than that of $Li_2OHCl$ and 2 orders of magnitude higher than that of previously reported[26]. The in situ powder X-ray diffraction (PXRD) and differential scanning calorimetry (DSC) revealed the phase transition mechanism, and double confirmed the stabilizing effect of K-doping on the metastable cubic phase of antiperovskite. The PXRD, powder neutron diffraction (PND), and atomic pair distribution function (PDF) refinements indicated that the substitution of K$^+$ for $[Li_2OH]^+$ cluster led to the stabilization of the cubic phase and lattice contraction. Both bond-valence site energy (BVSE) and ab initio molecular dynamics (AIMD) calculations reveal the advantages of $(Li_2OH)_{0.99}K_{0.01}Cl$ over $Li_2OHCl$ in three-dimensional migration trajectories from the perspective of ionic transport mechanism. Furthermore, the Li‖Li symmetric cell employing $(Li_2OH)_{0.99}K_{0.01}Cl$ exhibits stable cycling performance at 80 °C over 500 h, at current densities of 0.1 mA cm$^{-1}$ (with an areal capacity of 0.05 mAh cm$^{-1}$) and 0.2 mA cm$^{-1}$ (with an areal capacity of 0.1 mAh cm$^{-1}$). By combining $(Li_2OH)_{0.99}K_{0.01}Cl$ electrolyte with the excess Li-metal anode and the LiFePO$_4$ (LFP) cathode (loading of 1.78 mg cm$^{-2}$), we report an ASSLB achieving a specific capacity of 116.4 mAh g$^{-1}$ with a capacity retention of 96.1% by the 150th cycle when operated at 80 mA g$^{-1}$ and 120 °C.

## Results and discussion
### Structure determination
The phase structures of antiperovskite were distinctly transformed by K-doping (Fig. 1a), and the PXRD patterns of $Li_2OHCl$ and $(Li_2OH)_{1-x}K_xCl$ at RT (-25 °C) are presented in Fig. 1b. For $Li_2OHCl$, all peaks are assigned to the orthorhombic phase ($Pmc2_1$ space group) without any other distinct impurities. In contrast, the patterns of $(Li_2OH)_{1-x}K_xCl$

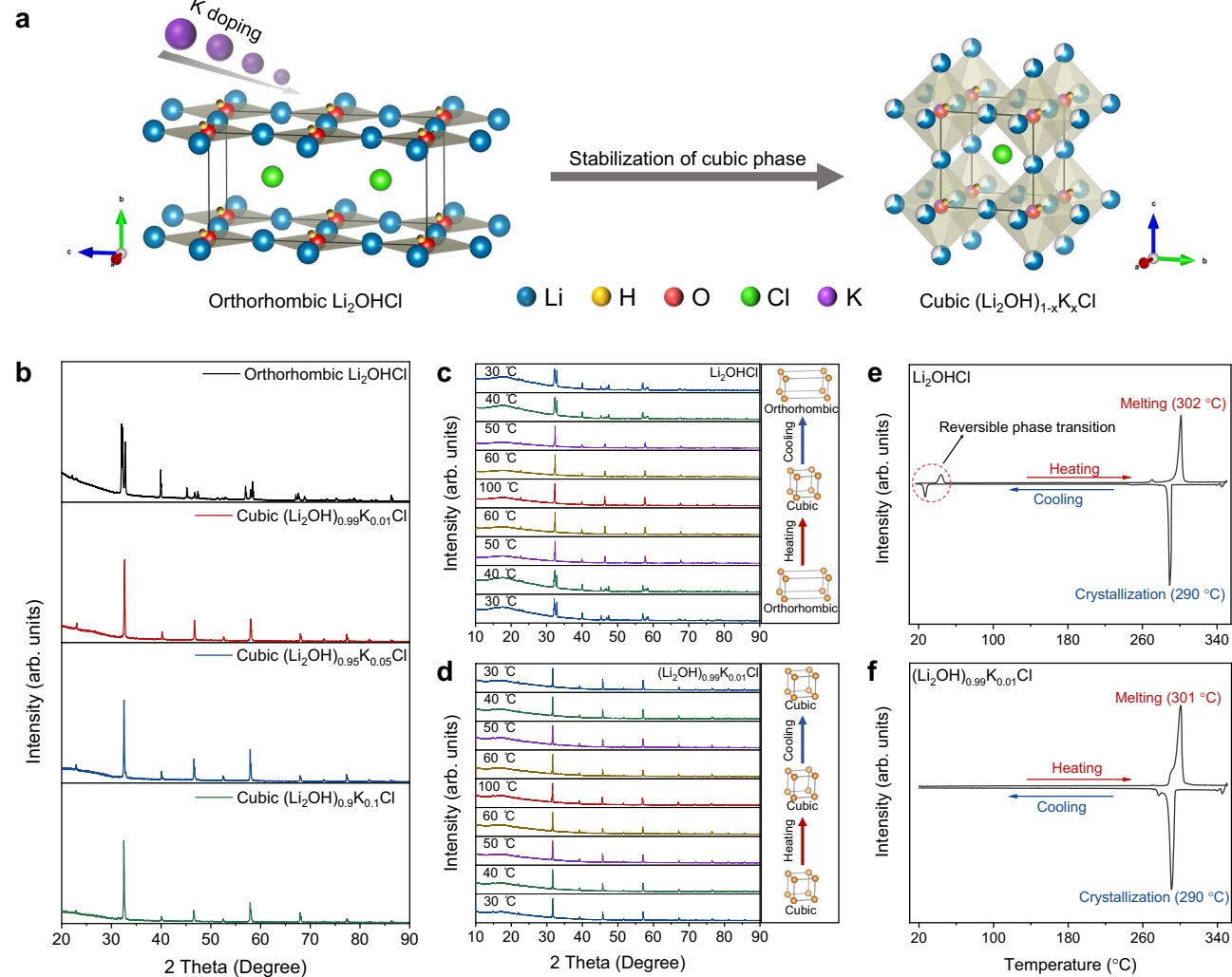

**Fig. 1 | Effect of K-doping on crystal phase of antiperovskite. a** Schematic diagram of cubic phase of antiperovskite stabilized by K-doping (Li in blue, O in red, H in yellow, Cl in green, K in purple). **b** PXRD patterns of orthorhombic $Li_2OHCl$ and cubic $(Li_2OH)_{1-x}K_xCl$ at RT (-25 °C). **c** In situ PXRD patterns of $Li_2OHCl$ in the range between 30 and 100 °C. **d** In situ PXRD patterns of $(Li_2OH)_{0.99}K_{0.01}Cl$ in the range between 30 and 100 °C. **e** DSC curve of $Li_2OHCl$. **f** DSC curve of $(Li_2OH)_{0.99}K_{0.01}Cl$.

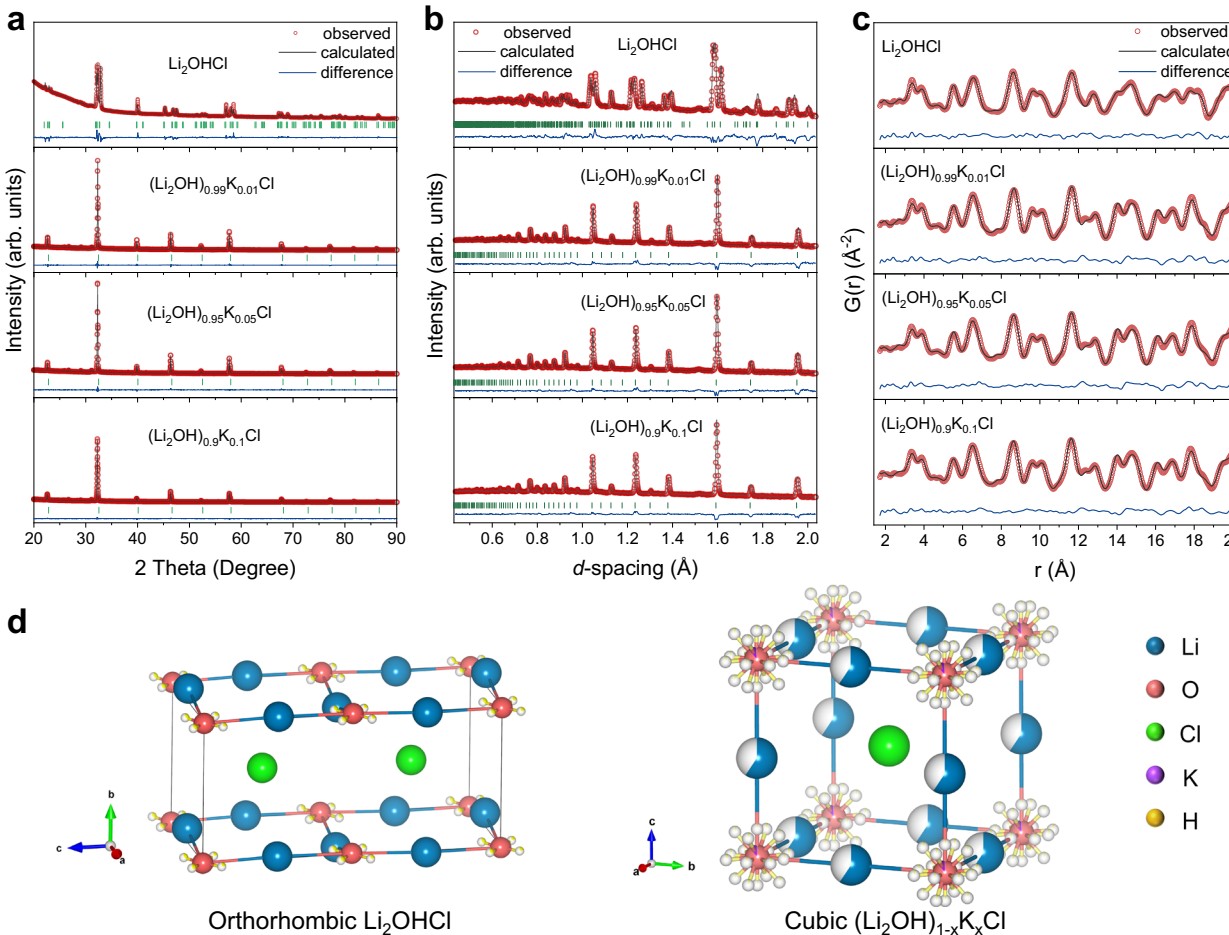

**Fig. 2 | Lattice structure analysis of Li₂OHCl and (Li₂OH)₁₋ₓKₓCl. a** The Rietveld refinements of PXRD data of orthorhombic Li₂OHCl and cubic (Li₂OH)₁₋ₓKₓCl. **b** The Rietveld refinements of PND data of orthorhombic Li₂OHCl and cubic (Li₂OH)₁₋ ₓKₓCl. **c** The small-box least-square refinements of PDF data of orthorhombic Li₂OHCl and cubic (Li₂OH)₁₋ₓKₓCl. **d** The crystal structure of orthorhombic Li₂OHCl and cubic (Li₂OH)₁₋ₓKₓCl according to the refinement results.

with K-doping exhibit distinct cubic phase ($Pm\bar{3}m$ space group) characteristics, which indicates that K-doping makes (Li₂OH)₁₋ₓKₓCl maintain the higher crystal symmetry. The in situ PXRD patterns of Li₂OHCl in the range between 30 and 100 °C are shown in Fig. 1c, which indicates that Li₂OHCl underwent a reversible phase transition between 40 and 50 °C. Taking the peaks at 30° to 35° 2θ as examples, the distinct three peaks (orthorhombic) turned into a sharp single peak (cubic), when the temperature rose from 40 to 50 °C. On the other hand, (Li₂OH)₀.₉₉K₀.₀₁Cl remains the same PXRD pattern characteristics of cubic phase in the range between 30 and 100 °C (Fig. 1d).

Also, the phase transition process is confirmed by the DSC results. As presented in Fig. 1e, the endothermic peak at 44 °C and exothermic peak at 27 °C corresponds to the reversible phase transition between orthorhombic phase and cubic phase of Li₂OHCl, and the endothermic peak at 302 °C and exothermic peak at 290 °C represent the melting and crystallization temperature, respectively. In contrast, except for melting and crystallization peaks, there are no other peaks for (Li₂OH)₀.₉₉K₀.₀₁Cl (Fig. 1f), which indicates that there is no phase transition from 20 °C to melting temperature. Besides, as the concentration of K-doping increases, the melting point decreases from 301 °C ((Li₂OH)₀.₉₉K₀.₀₁Cl) to 295 °C ((Li₂OH)₀.₉K₀.₁Cl) (Supplementary Fig. 1), which may be attributed to the fact that K⁺ ions disturbed the original ordering lattice arrangement in Li₂OHCl and reduced the lattice energy[33].

To gain an in-depth understanding of the relationship between K-doping and the lattice structures, the PXRD, PND, and PDF data were analyzed by structure refinements, respectively. As presented in Fig. 2a–c, all the PXRD, PND, and PDF data can be well simulated by the orthorhombic Li₂OHCl or cubic (Li₂OH)₁₋ₓKₓCl (see Supplementary Tables 1–12 for detailed crystallographic information). For orthorhombic Li₂OHCl, the crystal structure can be refined by the space group $PmC2_1$ (No. 26). The Li⁺ occupy two different Wyckoff positions (2a and 2b) and the OH⁻ ions are coordinated square-planarly by Li⁺ ions, which leads to a highly anisotropic and 2D structure (Figs. 1a, 2d). The H atoms occupy Wyckoff positions 4c around the O atom and have four possible crystallographic equivalent positions according to the PND data. For cubic (Li₂OH)₁₋ₓKₓCl, the lattices possess the higher symmetry and can be characterized by the space group $Pm\bar{3}m$ (No. 221). The OH⁻ ions are coordinated by Li⁺ ions, forming [Li₆OH] octahedra with one-third of Li vacancies for charge balance (Figs. 1a, 2d). The H atoms occupy Wyckoff positions 6e and 12i, and split into eighteen possible crystallographic equivalent positions according to the PND data. Unconventionally, as the foreign ions, the doping mechanism of K⁺ ions in cubic (Li₂OH)₁₋ₓKₓCl needs to be discussed carefully. Firstly, K and Li pertain to the congeners with similar chemical properties, so it seems logical to consider that K replaces the lattice site of Li and leads to the increase in cell size. Unexpectedly, with the increase of K doping (as shown in the EDS and XPS results of Fig. 3b, c), the lattice parameters obtained from PXRD and PND consistently decrease (Fig. 3a), which indicates that K may not occupy the Li sites. Also, the same interesting results are observed in (Li₂OH)₁₋ ₓKₓBr system (Supplementary Fig. 2). Secondly, it is assumed that K replaces Li in the antiperovskite lattice. The simulated PXRD according to the assumed structure model presents distinctly different

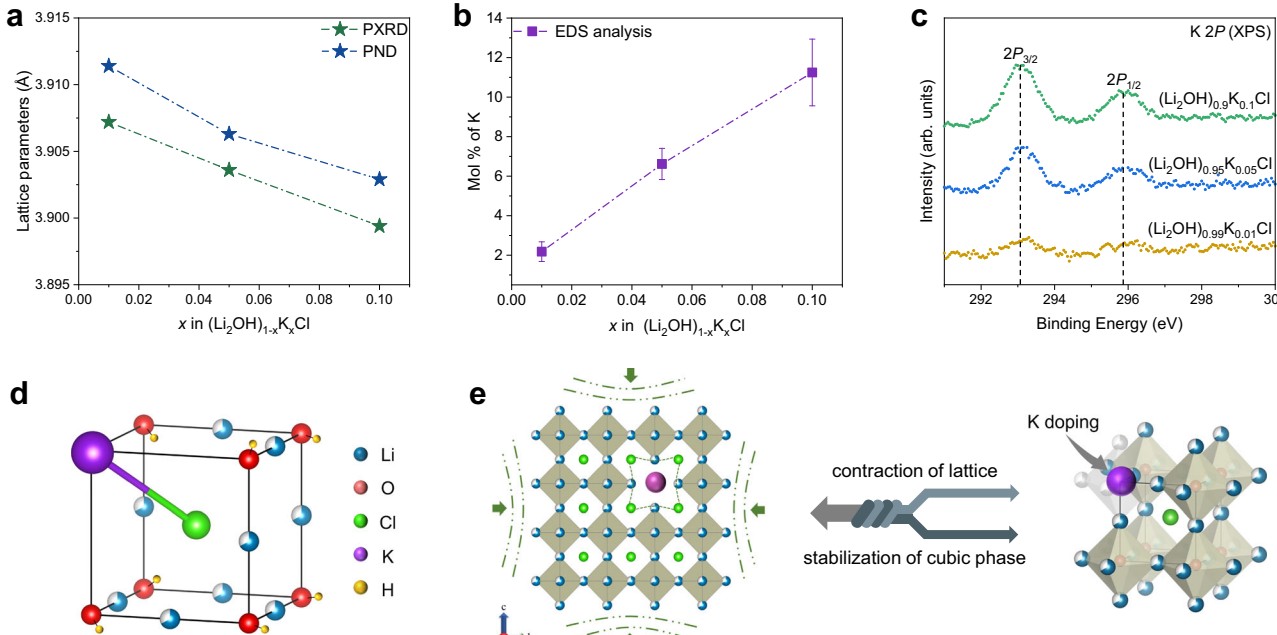

**Fig. 3 | The doping mechanism of K⁺ ions in antiperovskite (Li₂OH)₁₋ₓKₓCl. a** The lattice parameters of (Li₂OH)₁₋ₓKₓCl decreases with the increase of K-doping according to the PXRD and PND data. **b** The energy-dispersive X-ray spectroscopy (EDS) analysis of K in (Li₂OH)₁₋ₓKₓCl, and the error bars represent the standard deviation obtained by three independent measurements. **c** The X-ray photoelectron spectroscopy (XPS) of K 2P in (Li₂OH)₁₋ₓKₓCl. **d** The unit cell of cubic (Li₂OH)₁₋ₓKₓCl. **e** Illustration of lattice contraction and cubic phase stabilization induced by K-doping.

characteristics from the observed (e.g., the relative intensity of the diffraction peaks from 40° to 60°, Supplementary Fig. 3), which negates this hypothesis from the side. Thirdly, when refining the occupancy of O separately, it is always greater than 1.0, which indicates that there may be a larger atom at this site. Meanwhile, more Li vacancies emerge with the increase of K-doping according to the refinement results of PND. Hence, we infer that K⁺ ions may be located at (0, 0, 0) site and replace the [Li₂OH]⁺ cluster in [Li₆OH] octahedron (with one-third of Li vacancies), which leads to the disappearance of the original six coordination structure between OH⁻ and Li⁺ (Fig. 3d, e). It should be noted that the two unsubstituted Li⁺ of [Li₆OH] octahedron remain in the lattice and still coordinate with OH⁻ in other adjacent octahedra, and (Li₂OH)₁₋ₓKₓCl with K⁺ substituted [Li₂OH]⁺ cluster is charge balanced overall. Fourthly, the partial PDF data (Supplementary Figs. 4, 5) indicate that the nearest anion to K⁺ is Cl⁻, and it can be approximated that the coordination environment of K⁺ in (Li₂OH)₁₋ₓKₓCl is similar to that in KCl (space group: $Pm\bar{3}m$, Materials Project[34], No. mp-23289). Hence, the bonding mode between K⁺ and Cl⁻ also implies the formation of solid solution (Li₂OH)₁₋ₓKₓCl. Likewise, a similar mechanism has been reported in previous work, in which I⁻ ions substitute BH₄⁻ tetrahedral clusters and stabilize the metastable superionic phase of LiBH₄ at RT[35–38]. Fifthly, the tolerance factor (0.794) of Li₂OHCl is out of the stable range from 0.8 to 1.0 (Eq. 1) and close to the lower range value, where $R_{Li}$ is the radius of Li⁺ (0.76 Å), $R_{Cl}$ is the radius of Cl⁻ (1.81 Å), and $R_{OH}$ is the radius of OH⁻ (1.53 Å)[39,40]. While the radius of K⁺ (1.38 Å) is smaller than that of OH⁻ in (Li₂OH)₁₋ₓKₓCl, which contributes to the tolerance factor changing towards the stable range of cubic perovskite-type structure. Also, the formation energy ($E_f$) of (Li₂OH)₂₆KCl₂₇ supercell (3 × 3 × 3) is calculated to be −0.599 eV, indicating that the proposed structural model of K⁺ replacing [Li₂OH]⁺ cluster in (Li₂OH)₁₋ₓKₓCl is reasonable. Therefore, based on the analysis of PXRD, PND, and PDF data, it can be concluded that the doping of K⁺ ions in the antiperovskite lattice causes the unconventional double effects, that is, the stabilization of metastable cubic phase and the contraction of lattice (Fig. 3e). It is noteworthy that the doping mechanism involving K⁺ ions in (Li₂OH)₁₋ₓKₓCl significantly

differs from that of F⁻ ions in Li₂OHCl₀.₉F₀.₁ of our previously reported work[41], despite both lattice manipulation methods enabling the transformation of antiperovskite electrolytes from orthorhombic to cubic phase. In the case of Li₂OHCl₀.₉F₀.₁, F⁻ ions are conventionally introduced into the lattice to replace Cl⁻ ions. Whereas, K⁺ ions substitute for the [Li₂OH]⁺ clusters in an unconventional manner in (Li₂OH)₁₋ₓKₓCl lattice to form a solid solution.

$$t = (R_{Li} + R_{Cl})/\sqrt{2}(R_{Li} + R_{OH}) \qquad (1)$$

## Ionic conductivity

The ionic conductivities of (Li₂OH)₁₋ₓKₓCl electrolytes were assessed through temperature-dependent EIS. The Nyquist plots, as shown in Supplementary Fig. 6, indicate the current response of SSEs to a voltage disturbance at various frequencies and temperatures. Using the Nyquist curve of (Li₂OH)₀.₉₉K₀.₀₁Cl at 25 °C as an illustrative example (Supplementary Fig. 7), it presents a characteristic semicircle pattern at high frequencies indicating the series combination of bulk and grain boundary resistance, and the linear segment at low frequencies signifying ion blocking electrode. When fitting the plot using the equivalent circuit, it yields a total resistance ($R$) of 25.529 kΩ, comprising a bulk resistance ($R_b$) of 529 Ω and a grain boundary resistance ($R_{gb}$) of 25.0 kΩ. Subsequently, the ionic conductivity can be calculated as 4.5 × 10⁻³ mS cm⁻¹ using the formula $\sigma = D/(R\pi r^2)$, where $D$ (~0.9 mm), $R$ (~25.529 kΩ), $r$ (~0.5 cm) represent the thickness, total resistance, and radius of the (Li₂OH)₀.₉₉K₀.₀₁Cl electrolyte, respectively.

Figure 4a displays Arrhenius plots for the (Li₂OH)₁₋ₓKₓCl electrolytes indicating the correlation between ionic conductivity and temperature within the temperature range of 25 and 120 °C. For Li₂OHCl, after undergoing the phase transition from the orthorhombic to the cubic at 44 °C (Fig. 1c–e), both the ionic conductivity in the reported work[26] and this work exhibit a sharp increase. Noteworthily, due to the presence of LiCl impurity (about 30°, 35° 2θ) in Li₂OHCl previously

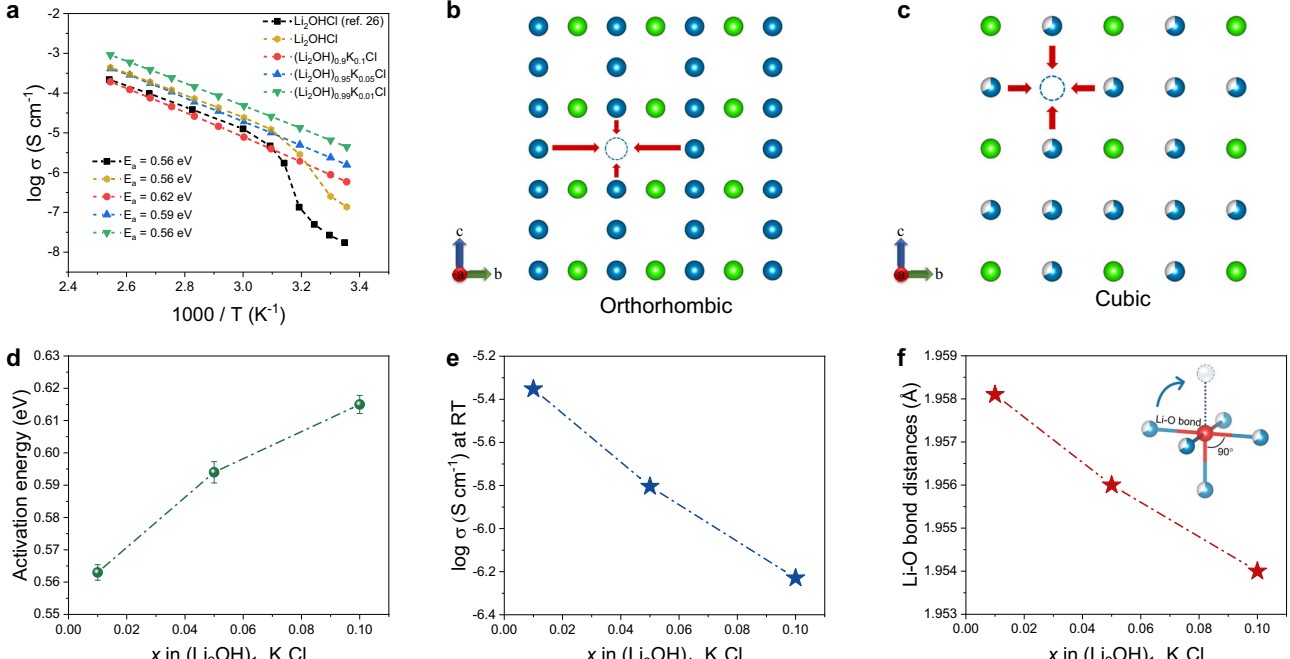

**Fig. 4 | Ionic conductivity analysis of Li₂OHCl and (Li₂OH)₁₋ₓKₓCl. a** The Arrhenius ionic conductivity plots of antiperovskite in the temperature range from 25 to 120 °C, the ionic conductivity data of Li₂OHCl (black) are reproduced from the work of Z. D. Hood et al.[26]. **b** Schematic diagram of nonequivalent sites for Li⁺ ions transport in the orthorhombic lattice. **c** Schematic diagram of equivalent sites for Li⁺ ions transport in the cubic lattice, Li⁺ ions in blue, Cl⁻ ions in green. **d** The

activation energy of (Li₂OH)₁₋ₓKₓCl increases with the increase of K-doping, and the error bars represent the standard deviation calculated by linearly fitting the Arrhenius ionic conductivity plots. **e** The ionic conductivity of (Li₂OH)₁₋ₓKₓCl decreases with the increase of K-doping. **f** The Li−O bond distance of (Li₂OH)₁₋ₓKₓCl decreases with the increase of K-doping according to the PDF data.

reported, its ionic conductivity is lower than in this work. For (Li₂OH)₁₋ₓKₓCl, because the cubic phase is always kept in the range from 25 to 120 °C, the ionic conductivities increase steadily and linearly with the increase in temperature. In particular, the ionic conductivity of (Li₂OH)₀.₉₉K₀.₀₁Cl is significantly increased to $4.5 \times 10^{-3}$ mS cm⁻¹ at 25 °C, which is 2 orders of magnitude higher than that of Li₂OHCl previously reported ($1.7 \times 10^{-5}$ mS cm⁻¹)[26] and 32 times higher than that of Li₂OHCl in this work ($1.37 \times 10^{-4}$ mS cm⁻¹). Besides, the activation energies of cubic Li₂OHCl and (Li₂OH)₁₋ₓKₓCl are obtained from the slope in the Arrhenius plots based on the formula: $\sigma = A \exp(-E_a/k_B T)$. In which $\sigma$ is the ionic conductivity, $A$ is the preexponential parameter, $E_a$ is the activation energy, $T$ is absolute temperature, and $k_B$ is the Boltzmann constant. Meanwhile, the activation energies increase as the concentration of K-doping increases, from 0.56 eV ((Li₂OH)₀.₉₉K₀.₀₁Cl) to 0.62 eV ((Li₂OH)₀.₉K₀.₁Cl).

The above ionic conductivity results could be understood from the perspective of lattice structure. Generally, the ionic conductivity of metastable phase is higher than that of stable phase. For example, the ionic conductivity of Li₇La₃Zr₂O₁₂ (LLZO) was remarkably improved after the metastable cubic phase was stabilized by Al/Ta doping at RT[31,32]. This is because, with the higher symmetry, the cubic LLZO lattice possesses more Li equivalent sites, which is beneficial for Li-ions to transport to multiple vacancy directions[42]. Moreover, lattice vacancy is another important factor affecting the transport of Li⁺ ions[8]. According to the crystallographic information (Supplementary Tables 1–12), the occupancy of Li1 and Li2 in orthorhombic Li₂OHCl is close to 1. In comparison, there are about one-third of Li vacancies in cubic (Li₂OH)₁₋ₓKₓCl, which is helpful for the transport of Li⁺ ions in the lattice. Therefore, with more Li equivalent sites and more Li vacancies (Fig. 4b, c), the ionic conductivity of (Li₂OH)₁₋ₓKₓCl is distinctly higher than that of Li₂OHCl at RT. On the other hand, a longer Li-anion bond usually means a wider diffusion pathway and a weaker interaction force between Li and anion[43]. According to the PDF data, the Li-O bond

length of (Li₂OH)₁₋ₓKₓCl increases with the decrease of K-doping (Fig. 4f). Hence, the lattice of (Li₂OH)₀.₉₉K₀.₀₁Cl is more favorable for the transport of Li⁺ ions with the lowest activation energy of 0.56 eV (Fig. 4d), leading to the highest ionic conductivity of $4.5 \times 10^{-3}$ mS cm⁻¹ at 25 °C (Fig. 4e). It should be noted that the ionic conductivity of (Li₂OH)₀.₉₉K₀.₀₁Cl is slightly higher than that of cubic Li₂OHCl after phase transition, which may be attributed to a small number of cluster defects caused by K-doping, and there may be a mutual restriction relationship between cluster defects and Li-O bond length in affecting ionic conductivity of (Li₂OH)₁₋ₓKₓCl.

Besides, the migration pathways of Li⁺ ions and associated energy barriers were calculated by the BVSE method. In the case of orthorhombic Li₂OHCl, as previously demonstrated in our studies[41], the [Li1-Li2-Li1] trajectory in $a$–$c$ plane has been identified as a favorable pathway for the transport of Li⁺ ions, featuring an effective barrier of 0.513 eV. However, the 3D long-range migration of Li⁺ ions along [Li1-Li1] trajectory requires overcoming a higher barrier of 0.832 eV. This finding underscores that the dominant mode of Li⁺ ions transport in Li₂OHCl occurs within the 2D plane. In contrast, for (Li₂OH)₀.₉₉K₀.₀₁Cl in this study (Fig. 5a–c), the isotropic [Li1-Li1] trajectory forms a 3D percolating network with an effective migration barrier of 0.326 eV, resulting in the much higher ionic conductivity. Indeed, the merits of the 3D transport pathway have been exemplified in the cubic Li₂OHCl₀.₉F₀.₁ structure in our prior study[41], which corresponds to the Li⁺ ions migration barrier of 0.439 eV. However, due to the stronger Li–F bonds introduced by F-doping compared to the original Li–Cl bonds, the barrier that needs to be overcome for Li⁺ ions migration along the [Li1-Li1] trajectory in Li₂OHCl₀.₉F₀.₁ (0.439 eV) is higher than that in (Li₂OH)₀.₉₉K₀.₀₁Cl (0.326 eV). Also, AIMD calculations embody the advantages of cubic K-doped antiperovskite in 3D migration trajectories. As shown in Fig. 5d, f, Li atoms trajectories in the orthorhombic Li₂OHCl show a local distribution around the lattice sites, and the MSD of Li atoms keeps steady with the simulation time. In contrast, Li atoms

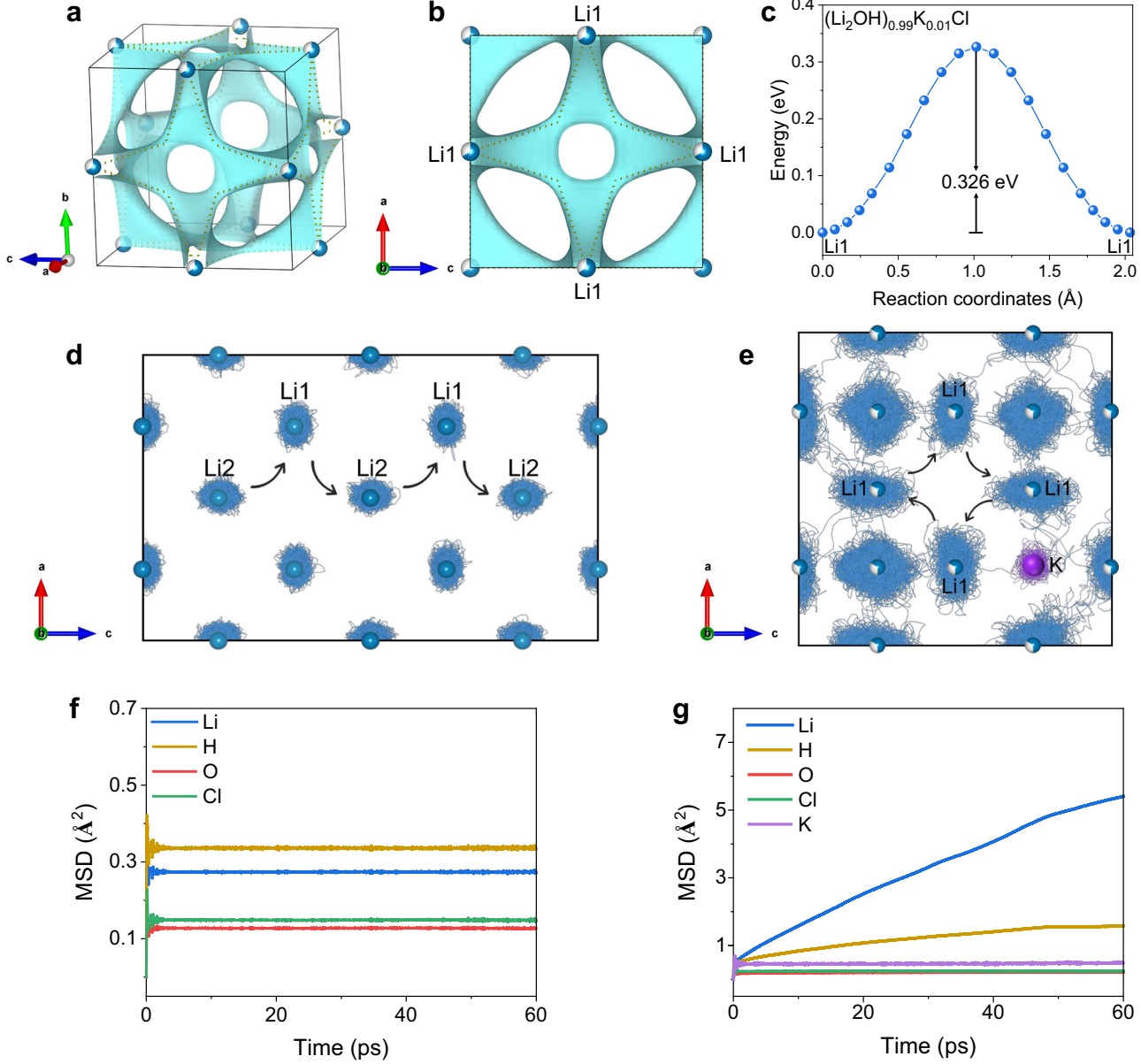

**Fig. 5 | Calculations of Li⁺ ions migration in the antiperovskite lattice. a, b** BVSE analysis for cubic $(Li_2OH)_{0.99}K_{0.01}Cl$ with the Li⁺ ions potential map, in which the dotted lines represent the topology of the Li⁺ ions migration pathways. **c** Energy profile of Li⁺ ions pathways in $(Li_2OH)_{0.99}K_{0.01}Cl$ as estimated from the BVSE models. **d, e** AIMD calculations: trajectories of Li atoms in orthorhombic lattice and cubic K-doped lattice. **f, g** MSD of different atoms in orthorhombic lattice and cubic K-doped lattice.

in cubic K-doped antiperovskite lattice exhibit significant dispersion characteristics and 3D migration trajectories (Fig. 5e), and the MSD of Li atoms is obviously higher than that of other atoms (Fig. 5g), which indicates that the ionic conductivity measured in experiments is attributed to Li⁺ diffusion. Coincidentally, the nuclear density maps derived from PND data using the maximum entropy method (MEM) also confirm the characteristics of the 3D transport path of Li⁺ ions in cubic $(Li_2OH)_{0.99}K_{0.01}Cl$ from experimental evidence (Supplementary Fig. 8).

### Electrochemical performance of $(Li_2OH)_{0.99}K_{0.01}Cl$

In addition to demonstrating the high ionic conductivity of $(Li_2OH)_{0.99}K_{0.01}Cl$, its electrochemical performance was systematically studied. As presented in Fig. 6a–c, the $Li|(Li_2OH)_{0.99}K_{0.01}Cl|Li$ symmetric cell remained stable after long-term galvanostatic cycling (current density of 0.1 mA cm⁻², areal capacity of 0.05 mAh cm⁻², 500 h; current density of 0.2 mA cm⁻², areal capacity of 0.1 mAh cm⁻², 500 h)

at 80 °C without any distinct polarization. Moreover, a critical current density (CCD) test was conducted on $(Li_2OH)_{0.99}K_{0.01}Cl$ to investigate the short-circuit tolerance (Fig. 6d). With the increase of the current density from 0.05 mA cm⁻² to 0.7 mA cm⁻², the cell voltage increased steadily from 12 mV to 173 mV.

After that, the cell voltage fluctuated unsteadily and dropped to a lower value, which indicates that $(Li_2OH)_{0.99}K_{0.01}Cl$ can tolerate a current density of 0.7 mA cm⁻² before short-circuit. Both cyclic voltammetry (CV) and linear sweep voltammetry (LSV) were carried out at 80 °C and the scan rate of 1 mV S⁻¹ to investigate the electrochemical stability of $(Li_2OH)_{0.99}K_{0.01}Cl$. For the CV test (Fig. 6e), the scanning curve is highly stable in the range of –0.5-3.5 V (vs. Li/Li⁺), indicating the $(Li_2OH)_{0.99}K_{0.01}Cl$ electrolyte is stable enough to resist reduction by Li metal. For the LSV test (Fig. 6f), the scanning curve shows that the current increases suddenly at 4.0 V (vs. Li/Li⁺), corresponding to the oxidative decomposition of $(Li_2OH)_{0.99}K_{0.01}Cl$. Therefore, the $(Li_2OH)_{0.99}K_{0.01}Cl$ electrolyte is stable in the voltage range of –0.5 to

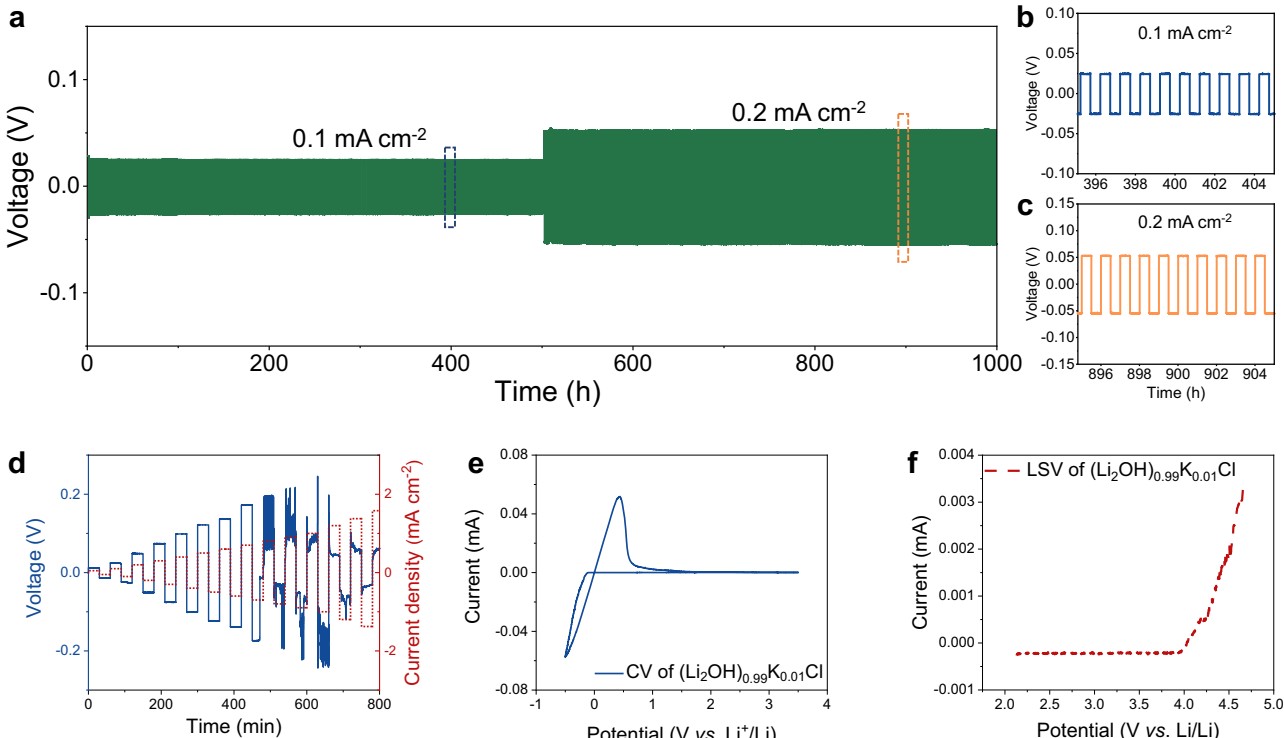

**Fig. 6 | The electrochemical performance of $(Li_2OH)_{0.99}K_{0.01}Cl$. a** The galvanostatic cycling of Li|$(Li_2OH)_{0.99}K_{0.01}$Cl|Li symmetric cell at the current density of 0.1 mA cm⁻² (areal capacity of 0.05 mAh cm⁻², 500 h) and 0.2 mA cm⁻² (areal capacity of 0.1 mAh cm⁻², 500 h). **b, c** Insets of the specific period of the galvanostatic cycling. **d** The critical current density (CCD) test for investigating the short-circuit tolerance of $(Li_2OH)_{0.99}K_{0.01}Cl$. The Li||Li symmetric cell was applied with a gradually increasing current density and was charged or discharged for 30 min at each current density during the CCD test. **e** The CV curve of $(Li_2OH)_{0.99}K_{0.01}Cl$ in the range of −0.5 to 3.5 V (versus Li/Li⁺) at the scanning rate of 1 mV s⁻¹. **f** The LSV curve of $(Li_2OH)_{0.99}K_{0.01}Cl$ at the scanning rate of 1 mV s⁻¹ and starting at the open-circuit voltage (~2.1 V) (versus Li/Li⁺). All of the above electrochemical tests were performed at 80 °C.

4.0 V (*vs.* Li/Li⁺) according to the results of CV and LSV, which suggests that it is suitable for ASSLBs with charging voltage below 4.0 V (e.g., Li| SSEs|LFP ASSLBs).

## Performance of Li|$(Li_2OH)_{0.99}K_{0.01}$Cl|LiFePO₄ ASSLB

As shown in Fig. 7a, b, the charge–discharge tests of Li| $(Li_2OH)_{0.99}K_{0.01}$Cl|LFP ASSLB present excellent stability, low polarization, and reversibility of redox reactions in a 2.9–3.8 V range. The cell exhibits a capacity retention of 96.1% by the 150th cycle with a specific capacity of 116.4 mAh g⁻¹ when operated at a specific current of 80 mA g⁻¹ and 120 °C, and there are no obvious changes in the shape of the voltage signatures (Fig. 7b). Overall, the ASSLB employing $(Li_2OH)_{0.99}K_{0.01}Cl$ electrolyte exhibits long-term stable cycling performance. More importantly, no liquid electrolyte or interface modification layer was introduced in any preparation process of the cell, in contrast to the majority of other solid-state battery studies[44–47].

Furthermore, the Li|SSE|LFP cell was disassembled to observe the volume distribution and the morphology of SSE-electrode interfaces after cycles. As presented in Fig. 7c, the three-dimensional images obtained by computed X-ray tomography (CT) scanning exhibit the integrity of Li|$(Li_2OH)_{0.99}K_{0.01}$Cl|LFP ASSLB in volume distribution, without structural damage and interface failure. Also, good adhesion can be observed in the transition region between $(Li_2OH)_{0.99}K_{0.01}Cl$ and Li metal anode (Fig. 7d), as well as the transition region between $(Li_2OH)_{0.99}K_{0.01}Cl$ and LFP cathode (Fig. 7e). In particular, the $(Li_2OH)_{0.99}K_{0.01}Cl$ bulk shows good uniformity and density without obvious porosity and graininess only by cold pressing and heating treatment at 120 °C. This rare advantage indicates that the low melting point of antiperovskite SSEs is favorable for its densification with low-energy consumption, unlike garnet[19] and NASICON[11] SSEs that need about 1000 °C sintering temperature to densify.

## Methods

### Material synthesis

The $(Li_2OH)_{1-x}K_xCl$ ($0 \leq x \leq 0.1$) were prepared by simply grinding stoichiometric amounts of LiOH (>99%), LiCl (>99%), and KOH(>99%) and one-step heating to 400 °C at the rate of 5 °C min⁻¹ for 1 h in a nickel crucible. Then the molten products were quenched to room temperature into another empty nickel crucible. All the above preparations were performed under Ar atmosphere.

### Material characterizations and analysis

PXRD measurements were conducted on a Rigaku SmartLab 9 kW diffractometer with Cu Kα radiation (λ = 1.5406 Å) to analyze the structures of $(Li_2OH)_{1-x}K_xCl$. All samples were sealed with Kapton film under the inert atmosphere before PXRD measurements. During the in situ PXRD testing in the temperature range of 30–100–30 °C, the samples were sealed in a vacuum chamber with an X-ray transparent beryllium window. The PND was measured at the Multiple Physics Instrument at the China Spallation Neutron Source (0.1–3 Å of wavelength range), and each sample was sealed in a Ti–Zr tube with a diameter of 8.9 mm. The Bank 6 PND data was selected for refinement because of its high resolution and wide *d*-spacing range (0.4–2.4 Å). DSC analysis was conducted on a NETZSCH STA 449 instrument, and samples were sealed in aluminum crucible under N₂ atmosphere with a heating and cooling rate of 5 °C/min. X-ray photoelectron spectroscopy (XPS) was recorded on a PHI 5000 VersaProbe II spectrometer with Al Kα X-rays. Field emission scanning electron microscopy (SEM, Hitachi SU8230) was used to observe the cross-sectional morphology of the interface between SSE and electrode, which was equipped with a liquid-nitrogen-cooled silicon detector to perform the energy-dispersive X-ray spectroscopy (EDS). PXRD and PND Rietveld refinements were performed by the Fullprof software[48] to analyze the space

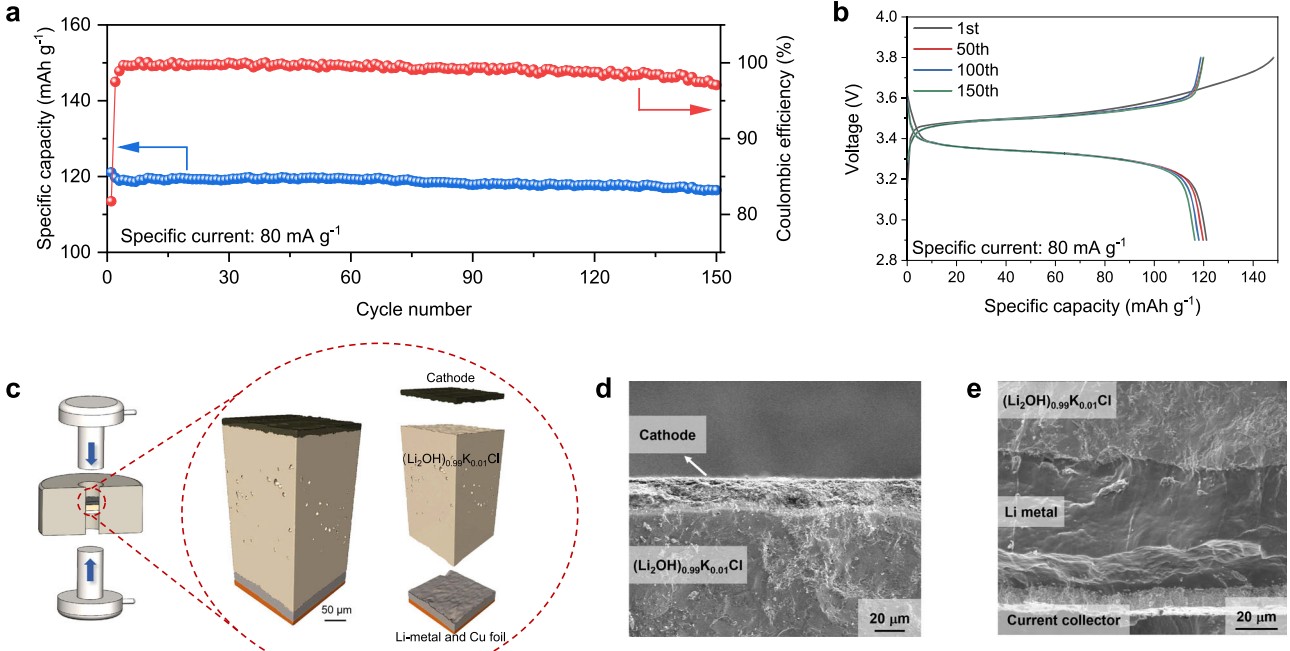

**Fig. 7 | The performance of Li|(Li$_2$OH)$_{0.99}$K$_{0.01}$Cl|LFP ASSLB. a** The cycling performance of Li|(Li$_2$OH)$_{0.99}$K$_{0.01}$Cl|LFP ASSLB at the specific current of 80 mA g$^{-1}$ (current density of 0.18 mA cm$^{-2}$) and 120 °C. **b** The voltage profiles of Li|(Li$_2$OH)$_{0.99}$K$_{0.01}$Cl|LFP ASSLB cycled between 2.9 and 3.8 V. **c** The three-dimensional images of Li|(Li$_2$OH)$_{0.99}$K$_{0.01}$Cl|LFP ASSLB obtained by CT scanning. **d** The cross-sectional morphology of the interface between LFP cathode and (Li$_2$OH)$_{0.99}$K$_{0.01}$Cl electrolyte observed by SEM. **e** The cross-sectional morphology of the interface between Li-metal anode and (Li$_2$OH)$_{0.99}$K$_{0.01}$Cl electrolyte observed by SEM.

group, crystal lattice parameters, and atom locations of antiperovskite Li$_2$OHCl and (Li$_2$OH)$_{1-x}$K$_x$Cl. In order to verify the PXRD results and confirm the effect of K-doping on the lattice structures of (Li$_2$OH)$_{1-x}$K$_x$Cl, the X-ray total scattering was performed on the PANalytical Empyrean powder diffractometer with Ag Kα radiation (0.559 Å) and GaliPIX$^{3D}$ detector at RT. The sample was sealed into a glass capillary with a diameter of 1.0 mm. The total scattering data were collected in the range from 2 to 140° in 17 h. The obtained coherent X-ray scattering data $I_C(Q)$ was converted into the structure-function $S(Q)$, and the structure-function $S(Q)$ was Fourier transformed into PDF with a $Q$ range of 0.4−20 Å$^{-1}$[49]. The PDF refinements were performed by PDFgui[50].

## Computational methods

BVSE calculations were used to simulate Li$^+$ ions migration pathways and evaluate migration barrier heights and performed with the *soft*BV program[51,52] using the structure models according to the Rietveld refinement data. Bond-valence site energy landscapes of a test Li$^+$ ion were calculated for a 3D grid of points with a resolution of 0.1 Å. Details about the *soft*BV approach and the type of force field employed for the BVSE calculations can be found elsewhere[51,52]. The AIMD calculations were carried out by using the projector augmented wave method in the framework of the density functional theory (DFT) as implemented in the Vienna ab-initio Simulation Package (VASP)[53] and performed by using the Verlet algorithm and the NVT ensemble with a Nosé−Hoover thermostat at 550 K[54]. A system contains 3 × 3 × 3 supercell ((Li$_2$OH)$_{26}$KCl$_{27}$) was used for the AIMD calculations of cubic K-doped antiperovskite, and the simulation cell for orthorhombic system was constructed from a 2 × 2 × 3 replication of the orthorhombic Li$_2$OHCl unit cell. The plane-wave energy cutoff of 520 eV was chosen for AIMD simulations. The time step was set to 1 fs, and all supercell systems were simulated for 60 ps after an initial 2 ps equilibration period, and a 1 × 1 × 1 $k$-mesh grid was used for the Brillouin-zone sampling. The mean-squared displacement (MSD) data were extracted from the AIMD trajectories of atoms. The maximum entropy method (MEM)

analysis was performed with the program Dysnomia[55] using an input file containing observed structure factors from the NPD data of Bank 6. Visualization of nuclear densities and extraction of 2D displays was then performed in the program Vesta[56]. The formation energy ($E_f$) is calculated from the energy difference between cubic Li$_2$OHCl and K-doped structure (Eq. 2) to demonstrate the validity of (Li$_2$OH)$_{1-x}$K$_x$Cl structure model.

$$E_f = E[(Li_2OH)_{26}Cl_{27}K] + 2 \times E[LiOH] - E[(Li_2OH)_{27}Cl_{27}] - E[KOH] \quad (2)$$

where $E[(Li_2OH)_{27}Cl_{27}]$, $E[(Li_2OH)_{26}KCl_{27}]$, $E[LiOH]$, and $E[KOH]$ represent the DFT energies of Li$_2$OHCl supercell (3 × 3 × 3), K-doped supercell (3 × 3 × 3), LiOH bulk phase and KOH bulk phase, respectively.

## Cell assembly and electrochemical measurements

Ionic conductivities in the temperature range of 25−120 °C were obtained from electrochemical impedance spectroscopy (EIS) at frequencies from 1 Hz to 1 MHz. Before the EIS measurements, Li$_2$OHCl and (Li$_2$OH)$_{1-x}$K$_x$Cl powders (~140 mg) were pressed into pellets by 480 MPa pressure in an insulative mold (inner diameter = 10 mm), and two stainless-steel rods were clamped on both sides of the pellet as the current collectors. The electrochemical stability window of (Li$_2$OH)$_{0.99}$K$_{0.01}$Cl was determined by cyclic voltammetry (CV) and linear sweep voltammetry (LSV) measurements. In the CV and LSV tests, the SSE pellets were sealed in 2032 coin-type cells, Li metal was used as the counter electrode and Au foil (diameter of 9 mm) was used as the working electrode. The CV was measured between –0.5 V and 3.5 V (versus Li/Li$^+$) at a scanning rate of 1 mV/s at 80 °C, and LSV was measured from open-circuit voltage (~2.1 V) to 5 V (versus Li/Li$^+$) at a scanning rate of 1 mV/s at 80 °C. All the above measurements were performed on an electrochemical workstation analyzer (AUTOLAB M204).

The Li|SSE|Li symmetric batteries were sealed in 2032 coin-type cells and then carried out at different current densities at 80 °C. Before

assembling the Li|SSE|Li symmetric cells, $(Li_2OH)_{0.99}K_{0.01}Cl$ powders (~100 mg) were pressed into SSE pellets in a stainless-steel mold (inner diameter = 9 mm), and then heated at 280 °C for 10 h for densification. The ASSLBs were assembled in polyether-ether-ketone (PEEK) molds by employing LFP as cathode, together with Li-metal anode and $(Li_2OH)_{0.99}K_{0.01}Cl$ SSE located between the cathode and the anode. For the cathode electrodes, the commercial LFP powders were mixed with the $Li_3InCl_6$ SSE powders (MTI Corp.) with a weight ratio of 70: 30 by grinding for 30 min. The ASSLBs were fabricated as follows: 70 mg of the $(Li_2OH)_{0.99}K_{0.01}Cl$ powders were pressed under 240 MPa to form an SSE layer with a thickness of around 400 μm. Then, 2 mg of cathode composite powders were spread onto the side of the SSE layer and pressed under 480 MPa. Finally, the Li foil (thickness of 100 μm, diameter of 9 mm and areal capacity of 20.6 mAh cm$^{-2}$) was attached to the other side of the SSE layer by pressing under 50 MPa to assemble the integrated ASSLB. After 12 h annealing at 120 °C for densification of SSE in the PEEK mold, the galvanostatic charge–discharge cycling of ASSLBs was carried out between 2.9 and 3.8 V at 120 °C. All the above measurements were conducted on the Wuhan Land battery tester.

## Reporting summary

Further information on research design is available in the Nature Portfolio Reporting Summary linked to this article.

## Data availability

The source data used in this study are available from Figshare (https://doi.org/10.6084/m9.figshare.22811984).

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

## Acknowledgements

This work was supported by the National Natural Science Foundation of China (51825201 and 52227802 to R.Z.), the National Natural Science Foundation of China (12275119 to S.H.), Guangdong Grants (2021ZT09C064 to S.H.), Shenzhen Science and Technology Program (KQTD20200820113047086 to J.Z.), Major Science and Technology Infrastructure Project of Material Genome Big-science Facilities Platform supported by Municipal Development and Reform Commission of Shenzhen. The authors are very grateful to W.Y., H.C.C., and Y.G.X. of the Multiple Physics Instrument at the China Spallation Neutron Source for their help in the neutron diffraction measurement.

## Author contributions

R.Z. and S.H. conceived the research, L.G. synthesized the materials, analyzed the crystal structures, conducted the electrochemical measurements under the guidance of R.Z., S.L., and L.W., H.Z. performed the XRD refinement with assistance from S.G., X.Z. performed the calculations with assistance from J.Z., L.G. analyzed the PDF and PND data under the guidance of J.Z. and Y.W., and discussed the mechanism analysis with Y.Z. and D.H., R.Z., and S.H. directed the entire study, L.G. handled the writing of the manuscript with other co-authors.

## Competing interests

The authors declare no competing interests.
