## [Peer Review File · Nature Communications]

Reviewers' comments:

Reviewer #1 (Remarks to the Author):

This is an interesting work that the cubic phase of Li_2OHCl can be stabilized by 1% doping of K, and the resultant $(\text{Li}_2\text{OH})_{0.99}\text{K}_{0.01}\text{Cl}$ shows a dramatic increase in ionic conductivity. I think this work will be of broad interest to the solid state electrolyte community. However, I still have some concerns that need to be clarified.

1. The authors claim that the doping occurs through the replacement of $[\text{Li}_6/2\text{OH}]$ by K^+ . This is different from the conventional idea that a K^+ just replaces a Li^+ . The authors used LiBH_4 as an analog that I^- ions substitute $[\text{BH}_4]^-$ tetrahedral clusters. But LiBH_4 is different, because each $[\text{BH}_4]^-$ is an isolated anion. In anti perovskite, each $[\text{Li}_6/2\text{OH}]$ is shared with 6 neighbors. So the replace of this cluster by a K^+ ion will dramatically change the local coordination. There should be stronger evidence than the refinement to prove this doping mechanism is valid. Can single crystal diffraction be done?
2. I am not convinced of the authors' argument on why 1% doping shows the highest conductivity. Figure 3b shows only slight differences in activation energy (from 0.56 eV to 0.59 eV and to 0.62 eV when doping changes from 1% to 5% and 10%). I am not use if the difference is in experimental error bar. Therefore, is the difference in ionic conductivity due to the difference in the change in bond length?
3. Also considering K^+ may be mobile too, especially if they are located in the large cavity by replacing the proposed $[\text{Li}_6/2\text{OH}]$ clusters. So is the ionic conductivity solely from Li^+ ions? Any contribution from K^+ ?
4. Figure 1e, how about the cooling? Does it show any solid-solid phase transition?
5. Figure 3a, the impedance data fitting does not make sense based on my study of Li_2OHBr and other people study.
6. For ionic conductivity, when it gets to the cubic phase (above 44 degree C), the best sample of K doped is only around two times higher than Li_2OHCl , which is not too promising.
7. Line 162 and abstract, authors should only compare samples prepared using the same method.
8. Figure 4e, the small oxidation peaks indicate the low Columbic efficiency for Li plating/stripping.
9. For the battery performance, reviewer cannot find the mass of cathode active material.

Reviewer #2 (Remarks to the Author):

In this paper, the authors report that the doping of potassium (K) to Li₂OHCl, an inverse perovskite compound that has attracted attention as a solid electrolyte for Li-ion secondary batteries, can suppress phase transition from a cubic to an orthorhombic phase during cooling temperatures and can realize high ionic conductivity of $4.5 \times 10^{-3} \text{ mS cm}^{-1}$ at a room temperature. Furthermore, they have confirmed the stability of the material to Li metal and demonstrated its operation as a solid-state battery. Although the reported physical property data is of some value, the analytical method to ensure the reliability of the data is judged to be insufficient. In particular, the novelty and the impact on the field are somewhat lacking for publication in Nature Communications, The paper is not accepted for publication. We recommend that the manuscript be submitted to a journal specialized in electrochemistry or solid state chemistry.

The following comments are made on the contents.

1. In the main text and Fig. 2, it is stated that K substitution leads to the lattice contraction and stabilizes the metastable cubic phase. Why is the cubic phase stabilized by K substitution? In general, the stability of perovskite-type (including reverse perovskite-type) structures as the cubic phase has been discussed in terms of the tolerance factors. From this point of view, the cubic crystal stabilization behavior of K-doped Li₂OHCl should be investigated once again.

2. In the (Li₂OH)_(1-x)K_xCl system, the lattice constant decreases as the K addition increases from 0.01 to 0.05 and then to 0.1. However, the lattice volume per chemical formula (V) for undoped Li₂OHCl in Supplementary Table 1 is $V = 3.8739 \times 3.8251 \times 7.9997 / 2 = 59.270 \text{ \AA}^3$ and for (Li₂OH)_{0.99}K_{0.01} the lattice volume V per chemical formula is $V = 3.9072^3 = 59.648 \text{ \AA}^3$ for (Li₂OH)_{0.99}K_{0.01}Cl in Supplementary Table 2, and is larger for (Li₂OH)_{0.99}K_{0.01}Cl. The lattice volume increases with the small amount of K addition at $x = 0.01$ but the volume decreases when concentrations of K increase. Why the lattice volume of Li₂OHCl show such complex behavior according to K doping?

3. the solid solution of K in the lattice of Li₂OHCl should be confirmed by some experimental technique such as ICP, SEM-EDS, TEM-EDS, etc.

4. Evaluation of activation energy by BVSE is effective for screening etc. but is difficult to reproduce quantitatively. The energy barriers to migration should be discussed taking into account atomic/electron level interactions such as first-principles calculations.

Reviewers' comments:

Reviewer #1 (Remarks to the Author):

This is an interesting work that the cubic phase of Li_2OHCl can be stabilized by 1% doping of K, and the resultant $(\text{Li}_2\text{OH})_{0.99}\text{K}_{0.1}\text{Cl}$ shows a dramatic increase in ionic conductivity. I think this work will be of broad interest to the solid state electrolyte community. However, I still have some concerns that need to be clarified.

Reply:

Thank you very much for your comment and giving us a valuable opportunity to make the manuscript improved. We have revised the manuscript according to your kind suggestion.

1. The authors claim that the doping occurs through the replacement of $[\text{Li}_6/2\text{OH}]$ by K^+ . This is different from the conventional idea that a K^+ just replaces a Li^+ . The authors used LiBH_4 as an analog that I⁻ ions substitute $[\text{BH}_4]^-$ tetrahedral clusters. But LiBH_4 is different, because each $[\text{BH}_4]^-$ is an isolated anion. In antiperovskite, each $[\text{Li}_6/2\text{OH}]$ is shared with 6 neighbors. So the replace of this cluster by a K^+ ion will dramatically change the local coordination. There should be stronger evidence than the refinement to prove this doping mechanism is valid. Can single crystal diffraction be done?

Reply:

Thank you for your suggestion. We agree with your view that the I⁻ ions substitute $[\text{BH}_4]^-$ clusters is different from the replacement of $[\text{Li}_6\text{OH}]^+$ octahedra by K^+ in this manuscript, and single crystal diffraction will be a very powerful means to prove this doping mechanism. Unfortunately, because $(\text{Li}_2\text{OH})_{1-x}\text{K}_x\text{Cl}$ solid electrolytes is extremely sensitive to humidity, its crystal growth and single crystal diffraction analysis are so challenging for us that it cannot be carried out at current stage. However, based on the analysis of powder X-ray diffraction (PXRD), powder neutron diffraction (PND, more accurate for analysis of Li, O, H atoms), and atomic pair distribution function (PDF) data, the doping mechanism can be proved to be reasonable in this work (**Fig. 2** in manuscript and **Table S1-12** in supplementary information).

Firstly, K and Li pertain to the congeners with similar chemical properties, so it seems logical to consider that K replaces the lattice site of Li and leads to the increase in cell size. Unexpectedly, with the increase of K doping (as shown in the EDS and XPS results of **Fig. 1**), the lattice parameters obtained from PXRD and PND consistently decrease, which indicates that K may not occupy the Li sites. Also, the same interesting results are observed in $(\text{Li}_2\text{OH})_{1-x}\text{K}_x\text{Br}$ system (**Fig. 2**). Secondly, it is assumed that K replaces Li in the antiperovskite lattice. The simulated PXRD according to the assumed structure model presents distinctly different characteristics from the observed (e.g., the relative intensity of the diffraction peaks from 40° to 60° , **Fig. 3**), which negates this hypothesis from the side. Thirdly, when refining the occupancy of O separately, it is always greater than 1.0, which indicates that there may be a larger atom at this site. Meanwhile, more Li vacancies emerge with the increase of K-doping according to the refinement results of PND. Hence, we infer that K^+ should be located at (0, 0, 0) site of O atom in the cubic lattice and substitute the $[\text{Li}_6\text{OH}]^+$ octahedral cluster. Fourthly, both KCl and Li_2OHCl have been experimentally reported to have $Pm\bar{3}m$ phase [1-3], which also implies the formation of solid solution $(\text{Li}_2\text{OH})_{1-x}\text{K}_x\text{Cl}$. Fifthly, the tolerance factor (0.794)

of Li_2OHCl is out of the stable range from 0.8 to 1.0 (Equation 1) and close to the lower range value, where R_{Li} is the radius of Li^+ (0.76 Å), R_{Cl} is the radius of Cl^- (1.81 Å), and R_{OH} is the radius of OH^- (1.53 Å). While the radius of K^+ (1.38 Å) is smaller than that of OH^- in $(\text{Li}_2\text{OH})_{1-x}\text{K}_x\text{Cl}$, which contributes to the tolerance factor changing towards the stable range of cubic perovskite-type structure. Therefore, based on the analysis of PXRD, PND, and PDF data, it can be concluded that the doping of K^+ ions in the antiperovskite lattice causes the unconventional double effects, that is, the stabilization of metastable cubic phase and the contraction of lattice.

Fig. 1 a The lattice parameters of $(\text{Li}_2\text{OH})_{1-x}\text{K}_x\text{Cl}$ decreases with the increase of K-doping according to the PXRD and PND data. b The EDS analysis of K in $(\text{Li}_2\text{OH})_{1-x}\text{K}_x\text{Cl}$. c The XPS of K 2P in $(\text{Li}_2\text{OH})_{1-x}\text{K}_x\text{Cl}$.

Fig. 2 The lattice parameters of $(\text{Li}_2\text{OH})_{1-x}\text{K}_x\text{Br}$ ($0 \leq x \leq 0.1$) decreases with the increase of K-doping according to the PXRD.

Fig. 3 a The simulated PXRD according to the assumed structure model of K substitution for Li sites in antiperovskite lattice. b The observed PXRD for comparison.

- [1] Hood ZD, Wang H, Samuthira Pandian A, Keum JK, Liang C. Li_2OHCl Crystalline Electrolyte for Stable Metallic Lithium Anodes. *J Am Chem Soc* **138**, 1768-1771 (2016).
- [2] Weir CE, Piermarini GJ. Lattice Parameters and Lattice Energies of High-Pressure Polymorphs of Some Alkali Halides. *Journal of Research of the National Bureau of Standards Section A: Physics and Chemistry* **68A**, 105 (1964).
- [3] Pravica M, Bai L, Bhattacharya N. High-pressure X-ray diffraction studies of potassium chlorate. *Journal of Applied Crystallography* **45**, 48-52 (2012).

Now we have added neutron diffraction data to the manuscript and inserted the discussion of the doping mechanism into the third paragraph in **Structure Determination** section as follows:

“Unconventionally, as the foreign ions, the doping mechanism of K^+ ions in cubic $(\text{Li}_2\text{OH})_{1-x}\text{K}_x\text{Cl}$ needs to be discussed carefully. Firstly, K and Li pertain to the congeners with similar chemical properties, so it seems logical to consider that K replaces the lattice site of Li and leads to the increase in cell size. Unexpectedly, with the increase of K doping (as shown in the EDS and XPS results of **Fig. 3b** and c), the lattice parameters obtained from PXRD and PND consistently decrease (**Fig. 3a**), which indicates that K may not occupy the Li sites. Also, the same interesting results are observed in $(\text{Li}_2\text{OH})_{1-x}\text{K}_x\text{Br}$ system (Supplementary Fig. S2). Secondly, it is assumed that K replaces Li in the antiperovskite lattice. The simulated PXRD according to the assumed structure model presents distinctly different characteristics from the observed (e.g., the relative intensity of the diffraction peaks from 40° to 60° , Supplementary Fig. S3), which negates this hypothesis from the side. Thirdly, when refining the occupancy of O separately, it is always greater than 1.0, which indicates that there may be a larger atom at this site. Meanwhile, more Li vacancies emerge with the increase of K-doping according to the refinement results of PND. Hence, we infer that K^+ ions substitute OH^- ions at the center of the $[\text{Li}_6\text{OH}]^+$ octahedra, and Li at the vertex of octahedra should be eliminated accordingly considering the charge balance. In other words, as a larger ion, K^+ is located at (0, 0, 0) site in the cubic lattice and substitute the $[\text{Li}_6\text{OH}]^+$ octahedral cluster. Fourthly, both KCl and Li_2OHCl have been experimentally reported to have $Pm\bar{3}m$ phase, which also implies the formation of solid solution $(\text{Li}_2\text{OH})_{1-x}\text{K}_x\text{Cl}$. Likewise, a similar mechanism has been reported in previous work, in which Γ^- ions substitute BH_4^- tetrahedral clusters and stabilize the metastable superionic phase of LiBH_4 at RT. Fifthly, the tolerance factor (0.794) of Li_2OHCl is out of the stable range from 0.8 to 1.0 (Equation 1) and close to the lower range value, where R_{Li} is the radius of Li^+ (0.76 Å), R_{Cl} is the radius of Cl^- (1.81 Å), and R_{OH} is the radius of OH^- (1.53 Å). While the radius of K^+ (1.38 Å) is smaller than that of OH^- in $(\text{Li}_2\text{OH})_{1-x}\text{K}_x\text{Cl}$, which contributes to the tolerance factor changing towards the stable range of cubic perovskite-type structure. Therefore, based on the analysis of PXRD, PND, and PDF data, it can be concluded that the doping of K^+ ions in the antiperovskite lattice causes the unconventional double effects, that is, the stabilization of metastable cubic phase and the contraction of lattice (**Fig. 3d**).”

2. I am not convinced of the authors' argument on why 1% doping shows the highest conductivity. Figure 3b shows only slight differences in activation energy (from 0.56 eV to 0.59 eV and to 0.62 eV when doping changes from 1% to 5% and 10%). I am not use if the

difference is in experimental error bar. Therefore, is the difference in ionic conductivity due to the difference in the change in bond length?

Reply:

Thank you for pointing it out, we apologize for the lack of activation energy error bar in the manuscript. Now we have performed linear fitting on the Arrhenius plots of $(\text{Li}_2\text{OH})_{1-x}\text{K}_x\text{Cl}$ and obtained the corresponding activation energy and error bar, as shown in **Fig. 4a** and **Table 1**. The error does not affect the variation trend of activation energy of $(\text{Li}_2\text{OH})_{1-x}\text{K}_x\text{Cl}$ with the increase of K-doping, and the activation energy and ionic conductivity are significantly related to K-doping. In fact, the change of activation energy is consistent with the change of Li–O bond length in principle. The lithium-ion migration is usually mainly affected by the Coulombic interaction (ionic bond energy) between lithium cation and its adjacent anions, specifically depending on the cations–anions bond length. The longer Li–O bond means the weaker Coulombic interaction between lithium cation and its adjacent oxygen anions, which leads to the lower lithium-ion migration barrier (activation energy) [4-6]. Further, the migration barrier (activation energy) affects the ionic conductivity. Hence, the structure of $(\text{Li}_2\text{OH})_{0.99}\text{K}_{0.01}\text{Cl}$ with longest Li–O bond is more favorable for the transport of Li^+ ions with the lowest activation energy of 0.56 eV, resulting in the highest ionic conductivity at RT.

Fig. 4 a The activation energy of $(\text{Li}_2\text{OH})_{1-x}\text{K}_x\text{Cl}$ increases with the increase of K-doping. **b** The ionic conductivity of $(\text{Li}_2\text{OH})_{1-x}\text{K}_x\text{Cl}$ decreases with the increase of K-doping. **c** The Li-O bond distance of $(\text{Li}_2\text{OH})_{1-x}\text{K}_x\text{Cl}$ decreases with the increase of K-doping according to the PDF data.

Table 1. The activation energy of cubic antiperovskite $(\text{Li}_2\text{OH})_{1-x}\text{K}_x\text{Cl}$.

x in $(\text{Li}_2\text{OH})_{1-x}\text{K}_x\text{Cl}$	Activation energy (eV)	Error
0.01	0.563	0.24%
0.05	0.594	0.33%
0.1	0.615	0.28%

[4] Abakumov AM, Fedotov SS, Antipov EV, Tarascon J-M. Solid state chemistry for developing better metal-ion batteries. *Nat Commun* **11**, 4976 (2020).

[5] Xu Z-M, Bo S-H, Zhu H. LiCrS_2 and LiMnS_2 cathodes with extraordinary mixed electron–ion conductivities and favorable interfacial compatibilities with sulfide electrolyte. *Appl Mater Interfaces* **10**, 36941-36953 (2018).

[6] Xu Z, Chen R, Zhu H. Li_2CuPS_4 superionic conductor: a new sulfide-based solid-state electrolyte. *J Mater Chem A* **7**, 12645-12653 (2019).

3. Also considering K⁺ may be mobile too, especially if they are located in the large cavity by replacing the proposed [Li₆/2OH] clusters. So is the ionic conductivity solely from Li⁺ ions? Any contribution from K⁺?

Reply:

Thank you for your question. For solid ionic conductors, ion transport is generally induced by the hopping between lattice sites. As the foreign ions, K⁺ ions are isolated in the lattice and not at the hopping sites. Hence K⁺ ions cannot transport between the lattice sites. In addition, the migrating ion size and available free space in the lattice are two intuitive factors that affect ions transport. The larger migrating ion size is more unfavorable for ion conduction in the lattice with the higher migration barrier [7]. The radius of K⁺ (1.38 Å) is obviously larger than that of Li⁺ (0.76 Å) [8], so K⁺ ions seem unlikely to transport with limited free space in the crystal lattice of (Li₂OH)_{1-x}K_xCl. Also, we performed the *ab-initio* molecular dynamics simulations on a system contains 3 × 3 × 3 supercell ((Li₂OH)₂₆KCl₂₇). Both mean-squared displacement (MSD) and atomic trajectories show that K⁺ ions are localized at lattice sites (**Fig. 5**), which indicates that there is no contribution from K⁺ to ionic conductivity.

Fig. 5 **a** MSD of different atoms in cubic K-doped antiperovskite lattice. **b** Trajectories of Li and K atoms in cubic K-doped antiperovskite lattice.

[7] Bachman JC, *et al.* Inorganic Solid-State Electrolytes for Lithium Batteries: Mechanisms and Properties Governing Ion Conduction. *Chem Rev* **116**, 140-162 (2016).

[8] Ahrens LH. The use of ionization potentials Part 1. Ionic radii of the elements. *Geochim Cosmochim Acta* **2**, 155-169 (1952).

4. Figure 1e, how about the cooling? Does it show any solid-solid phase transition?

Reply:

Thank you for your question. The cooling process of DSC (especially below RT) is important for explaining the mechanism of solid-solid phase transition. Now we have retested the DSC of Li₂OHCl and (Li₂OH)_{0.99}K_{0.01}Cl according to your question, especially in the range below RT during cooling process. As presented in **Fig. 6a**, the endothermic peak at 44 °C and exothermic peak at 27 °C corresponds to the reversible phase transition between orthorhombic phase and cubic phase of Li₂OHCl, and the endothermic peak at 302 °C and exothermic peak at 290 °C represent the melting and crystallization temperature, respectively. In contrast, except for melting and crystallization peaks, there are no other peaks for (Li₂OH)_{0.99}K_{0.01}Cl (**Fig. 6b**), which indicates that there is no solid-solid phase transition during the heating and cooling process.

Now we have replaced the DSC results of **Fig. 1e** and **Fig. 1f** in the manuscript, and described the corresponding contents in second paragraph of **Structure Determination** section as follows:

“Also, the above phase transition process is confirmed by the DSC results. As presented in **Fig. 1e**, the endothermic peak at 44 °C and exothermic peak at 27 °C corresponds to the reversible phase transition between orthorhombic phase and cubic phase of Li_2OHCl , and the endothermic peak at 302 °C and exothermic peak at 290 °C represent the melting and crystallization temperature, respectively. In contrast, except for melting and crystallization peaks, there are no other peaks for $(\text{Li}_2\text{OH})_{0.99}\text{K}_{0.01}\text{Cl}$ (**Fig. 1f**), which indicates that there is no phase transition from 20 °C to melting temperature.”

Fig. 6 a DSC curve of Li_2OHCl . **b** DSC curve of $(\text{Li}_2\text{OH})_{0.99}\text{K}_{0.01}\text{Cl}$.

5. Figure 3a, the impedance data fitting does not make sense based on my study of Li_2OHBr and other people study.

Reply:

Thank you for your suggestion. Now we have moved the results of impedance data fitting to the **Supplementary Materials**.

6. For ionic conductivity, when it gets to the cubic phase (above 44 degree C), the best sample of K doped is only around two times higher than Li_2OHCl , which is not too promising.

Reply:

Thank you for your comments. We agree with you that when Li_2OHCl is transformed into cubic phase, the ionic conductivity of K-doped antiperovskite SSE does not show significant advancement compared with Li_2OHCl . In fact, the metastable phase formed by doping has no significant advantage over the higher symmetry structure after phase transformation in other SSEs systems. For example, Ta-doped cubic $\text{Li}_{7-x}\text{La}_3\text{Zr}_{2-x}\text{Ta}_x\text{O}_{12}$ (LLZTO) has no advantage in ionic conductivity compared with $\text{Li}_7\text{La}_3\text{Zr}_2\text{O}_{12}$ (LLZO) which transforms into cubic phase at around 600 °C [9], as well as $\text{LiBH}_4\text{-LiI}$ solid solution which transforms into hexagonal phase at 115 °C (**Fig. 7**) [10]. On the other hand, the essence and novelty of this research work mainly focus on the unconventional effect caused by K doping, that is, K^+ ions substituting for $[\text{Li}_6\text{OH}]^+$ octahedral clusters in antiperovskite $(\text{Li}_2\text{OH})_{1-x}\text{K}_x\text{Cl}$, stabilizing the metastable cubic phase and the abnormal lattice contraction.

[9] Chen F, Li J, Huang Z, Yang Y, Shen Q, Zhang L. Origin of the Phase Transition in Lithium Garnets. *J Phys Chem C* **122**, 1963-1972 (2018).

[10] Maekawa H, *et al.* Halide-Stabilized LiBH_4 , a Room-Temperature Lithium Fast-Ion Conductor. *J Am Chem Soc* **131**, 894-895 (2009).

Fig. 7 **a** Arrhenius plot of the self-diffusivity of Ta-doped LLZO as a function of Ta dopant concentration [9]. **b** Ionic conductivities of LiBH₄ and LiBH₄-LiX [10].

7. Line 162 and abstract, authors should only compare samples prepared using the same method.
Reply:

Thank you for your suggestion. Now we have compared the ionic conductivity of Li₂OHCl and (Li₂OH)_{1-x}K_xCl prepared using the same method in this manuscript, and we have rewritten the relevant contents in second paragraph of **Ionic Conductivity** section as follows:

“In particular, the ionic conductivity of (Li₂OH)_{0.99}K_{0.01}Cl is significantly increased to 4.5 × 10⁻³ mS cm⁻¹ at RT, which is 2 orders of magnitude higher than that of Li₂OHCl previously reported (1.7 × 10⁻⁵ mS cm⁻¹) and 32 times higher than that of Li₂OHCl in this work (1.37 × 10⁻⁴ mS cm⁻¹).”

8. Figure 4e, the small oxidation peaks indicate the low Columbic efficiency for Li plating/stripping.
Reply:

Thank you for pointing it out. The small oxidation peak in the cyclic voltammetry (CV) curve may be caused by the poor contact between the working electrode and SSE during the test, which indicates the undesirable Li plating/stripping in the electrode/SSE interface. We performed the CV measurement again and ensured good interface contact between electrode and SSE during the test according to your suggestion. As presented in **Fig. 8**, the oxidation and reduction peaks of the CV curve show little difference in intensity, which indicates the high Columbic efficiency for Li plating/stripping. Meanwhile, the scanning curve is highly stable in the range of -0.5-3.5 V (vs. Li/Li⁺), indicating the (Li₂OH)_{0.99}K_{0.01}Cl SSE is stable enough to resist reduction by Li-metal and oxidation at 3.5 V by working electrode.

Fig. 8 The CV curve of (Li₂OH)_{0.99}K_{0.01}Cl in the range of -0.5-3.5 V (versus Li/Li⁺).

Now we have replaced the CV curve of $(\text{Li}_2\text{OH})_{0.99}\text{K}_{0.01}\text{Cl}$ in **Electrochemical Performance of $(\text{Li}_2\text{OH})_{0.99}\text{K}_{0.01}\text{Cl}$** section.

9. For the battery performance, reviewer cannot find the mass of cathode active material.

Reply:

Thank you for pointing it out, we apologize for the lack of the mass of cathode active material. Now we have clearly pointed out the mass of cathode active material in **Method** section as follows:

“70 mg of the $(\text{Li}_2\text{OH})_{0.99}\text{K}_{0.01}\text{Cl}$ powders were pressed under 240 MPa to form an SSE layer with a thickness of around 400 μm . Then, **2 mg of cathode composite powders** were spread onto the side of the SSE layer and pressed under 480 MPa. Finally, the Li foil was attached to the other side of the SSE layer by pressing under 50 MPa.”

Moreover, the performance of ASSLB has been improved by optimizing the battery preparation process, and the cell delivers a specific capacity of 116.4 mAh g^{-1} with a capacity retention of 96.1% after 150th cycle at the current density of 80 mA g^{-1} (**Fig. 9**).

Fig. 9 a The cycling performance of LFP | $(\text{Li}_2\text{OH})_{0.99}\text{K}_{0.01}\text{Cl}$ | Li ASSLB. **b** The Voltage profiles of LFP | $(\text{Li}_2\text{OH})_{0.99}\text{K}_{0.01}\text{Cl}$ | Li ASSLB at the current density of 80 mA g^{-1} .

Reviewer #2 (Remarks to the Author):

In this paper, the authors report that the doping of potassium (K) to Li₂OHCl, an inverse perovskite compound that has attracted attention as a solid electrolyte for Li-ion secondary batteries, can suppress phase transition from a cubic to an orthorhombic phase during cooling temperatures and can realize high ionic conductivity of 4.5×10^{-3} mS cm⁻¹ at a room temperature. Furthermore, they have confirmed the stability of the material to Li metal and demonstrated its operation as a solid-state battery. Although the reported physical property data is of some value, the analytical method to ensure the reliability of the data is judged to be insufficient. In particular, the novelty and the impact on the field are somewhat lacking for publication in Nature Communications, The paper is not accepted for publication. We recommend that the manuscript be submitted to a journal specialized in electrochemistry or solid state chemistry.

Reply:

Thank you very much for your comment and pointing out the shortcomings in the manuscript, so that we can improve the quality of this work. Now we have added more analytical methods to improve the reliability of this work according to your suggestion. Firstly, we have added the structural analysis method of powder neutron diffraction (PND) to further support the novel experimental phenomena and mechanism in the antiperovskite (Li₂OH)_{1-x}K_xCl system. Secondly, we performed SEM-EDS analysis on (Li₂OH)_{1-x}K_xCl and determined the content of potassium element. Thirdly, we carried out *ab-initio* molecular dynamics (AIMD) calculations and maximum entropy method (MEM) analysis to further elaborate the migration mechanism of Li⁺ ions in cubic K-doped antiperovskite lattice. Fourthly, we found a similar experimental phenomenon in the analogue (Li₂OH)_{1-x}K_xBr system, which confirms the conclusion about (Li₂OH)_{1-x}K_xCl from the side in this manuscript.

Overall, through analysis of neutron diffraction, X-ray diffraction and atomic pair distribution function data, we found for the first time that K⁺ replaced [Li₆OH]⁺ octahedral clusters, and significantly improved the ionic conductivity of (Li₂OH)_{0.99}K_{0.01}Cl at RT. As far as we know, there is no other phenomenon similar to K⁺ replacing cationic clusters in the field of solid-state electrolytes. We believe that the novel experimental phenomena and mechanism will arouse widespread interest in the field of solid electrolytes and solid-state batteries.

For responses to your comments/questions, please see below.

The following comments are made on the contents.

1. In the main text and Fig. 2, it is stated that K substitution leads to the lattice contraction and stabilizes the metastable cubic phase. Why is the cubic phase stabilized by K substitution? In general, the stability of perovskite-type (including reverse perovskite-type) structures as the cubic phase has been discussed in terms of the tolerance factors. From this point of view, the cubic crystal stabilization behavior of K-doped Li₂OHCl should be investigated once again.

Reply:

Thank you for your suggestion. We agree with you that the tolerance factor you mentioned is important for the discussion of stabilizing cubic perovskite-type structure. Indeed, K-doping

in Li_2OHCl tends to change the Goldschmidt tolerance factor to the range from 0.8 to 1.0 of stable cubic perovskite-type structure. According to the **Formula 1**, the tolerance factor (0.794) of Li_2OHCl is out of the stable range from 0.8 to 1.0 and close to the lower range value, where R_{Li} is the radius of Li^+ (0.76 Å), R_{Cl} is the radius of Cl^- (1.81 Å), and R_{OH} is the radius of OH^- (1.53 Å) [1, 2]. While K^+ is located at the position of OH^- in $(\text{Li}_2\text{OH})_{1-x}\text{K}_x\text{Cl}$ according to PND, PXRD and PDF data, and the radius of K^+ (1.38 Å) is smaller than that of OH^- , which contributes to the tolerance factor changing towards the stable range of cubic perovskite-type structure.

$$t = (R_{\text{Li}} + R_{\text{Cl}}) / \sqrt{2}(R_{\text{Li}} + R_{\text{OH}}) \quad (1)$$

- [1] Ahrens LH. The use of ionization potentials Part 1. Ionic radii of the elements. *Geochim Cosmochim Acta* **2**, 155-169 (1952).
 [2] Hagg G. Structural inorganic chemistry by A. F. Wells. *Acta Crystallogr* **15**, 921 (1962).

In addition, we have added neutron diffraction data to the manuscript and inserted the discussion of the doping mechanism into the third paragraph in **Structure Determination** section as follows:

“Unconventionally, as the foreign ions, the doping mechanism of K^+ ions in cubic $(\text{Li}_2\text{OH})_{1-x}\text{K}_x\text{Cl}$ needs to be discussed carefully. Firstly, K and Li pertain to the congeners with similar chemical properties, so it seems logical to consider that K replaces the lattice site of Li and leads to the increase in cell size. Unexpectedly, with the increase of K doping (as shown in the EDS and XPS results of **Fig. 3b** and **c**), the lattice parameters obtained from PXRD and PND consistently decrease (**Fig. 3a**), which indicates that K may not occupy the Li sites. Also, the same interesting results are observed in $(\text{Li}_2\text{OH})_{1-x}\text{K}_x\text{Br}$ system (Supplementary Fig. S2). Secondly, it is assumed that K replaces Li in the antiperovskite lattice. The simulated PXRD according to the assumed structure model presents distinctly different characteristics from the observed (e.g., the relative intensity of the diffraction peaks from 40° to 60° , Supplementary Fig. S3), which negates this hypothesis from the side. Thirdly, when refining the occupancy of O separately, it is always greater than 1.0, which indicates that there may be a larger atom at this site. Meanwhile, more Li vacancies emerge with the increase of K-doping according to the refinement results of PND. Hence, we infer that K^+ ions substitute OH^- ions at the center of the $[\text{Li}_6\text{OH}]^+$ octahedra, and Li at the vertex of octahedra should be eliminated accordingly considering the charge balance. In other words, as a larger ion, K^+ is located at (0, 0, 0) site in the cubic lattice and substitute the $[\text{Li}_6\text{OH}]^+$ octahedral cluster. Fourthly, both KCl and Li_2OHCl have been experimentally reported to have $Pm\bar{3}m$ phase, which also implies the formation of solid solution $(\text{Li}_2\text{OH})_{1-x}\text{K}_x\text{Cl}$. Likewise, a similar mechanism has been reported in previous work, in which I^- ions substitute BH_4^- tetrahedral clusters and stabilize the metastable superionic phase of LiBH_4 at RT. Fifthly, the tolerance factor (0.794) of Li_2OHCl is out of the stable range from 0.8 to 1.0 (Equation 1) and close to the lower range value, where R_{Li} is the radius of Li^+ (0.76 Å), R_{Cl} is the radius of Cl^- (1.81 Å), and R_{OH} is the radius of OH^- (1.53 Å). While the radius of K^+ (1.38 Å) is smaller than that of OH^- in $(\text{Li}_2\text{OH})_{1-x}\text{K}_x\text{Cl}$, which contributes to the tolerance factor changing towards the stable range of cubic perovskite-type structure. Therefore, based on the analysis of PXRD, PND, and PDF data, it can be concluded that the

doping of K^+ ions in the antiperovskite lattice causes the unconventional double effects, that is, the stabilization of metastable cubic phase and the contraction of lattice (**Fig. 3d**).”

2. In the $(Li_2OH)_{1-x}K_xCl$ system, the lattice constant decreases as the K addition increases from 0.01 to 0.05 and then to 0.1. However, the lattice volume per chemical formula (V) for undoped Li_2OHCl in Supplementary Table 1 is $V = 3.8739 \times 3.8251 \times 7.9997 / 2 = 59.270 \text{ \AA}^3$ and for $(Li_2OH)_{0.99}K$. The lattice volume V per chemical formula is $V = 3.9072^3 = 59.648 \text{ \AA}^3$ for $(Li_2OH)_{0.99}K_{0.01}Cl$ in Supplementary Table 2, and is larger for $(Li_2OH)_{0.99}K_{0.01}Cl$. The lattice volume increases with the small amount of K addition at $x = 0.01$ but the volume decreases when concentrations of K increase. Why the lattice volume of Li_2OHCl show such complex behavior according to K doping?

Reply:

Thank you for your question, we apologize for the ambiguous statement about the change of lattice volume in the manuscript. The change of lattice volume should be discussed in two aspects: (1) the phase transition between orthorhombic phase and cubic phase accompanied by the change of lattice volume. (2) the effect of K-doping concentration on the lattice volume of cubic $(Li_2OH)_{1-x}K_xCl$. The phase transition is usually accompanied by the rearrangement of atomic positions in the lattice, which has an impact on the lattice volume. For example, the abrupt volume expansion and sudden development of polarization can be induced by the phase transition of $PbZrO_3$ -based perovskite ceramics [3]. Also, the expansion of lattice volume occurs in azobenzene, VO_2 and ZrO_2 -based ceramics [4-6]. Likewise, it is reasonable that the lattice volume of cubic $(Li_2OH)_{1-x}K_xCl$ is larger than that of orthorhombic Li_2OHCl , because this phenomenon is mainly affected by phase transition. However, for the cubic $(Li_2OH)_{1-x}K_xCl$ system itself, the lattice volume of cubic $(Li_2OH)_{1-x}K_xCl$ gradually decreases with the increase of K-doping concentration, which is caused by the mechanism of K^+ replaced $[Li_6OH]^+$ octahedral clusters. In addition, the same results are observed in $(Li_2OH)_{1-x}K_xBr$ system. Interestingly, antiperovskite Li_2OHBr without K-doping is cubic at RT and does not involve K-doping induced phase transition. Therefore, when K^+ ions are introduced into cubic Li_2OHBr , its cell parameter decreases, and the cell parameters of $(Li_2OH)_{1-x}K_xBr$ decrease with the increase of K-doping concentration (**Fig. 1**). So overall, there is no contradiction in the change behaviors of lattice volume in this manuscript.

Fig. 1 The lattice parameters of $(Li_2OH)_{1-x}K_xBr$ ($0 \leq x \leq 0.1$) decreases with the increase of K-doping according to the PXRd.

- [3] Tan X, Ma C, Frederick J, Beckman S, Webber KG. The Antiferroelectric \leftrightarrow Ferroelectric Phase Transition in Lead-Containing and Lead-Free Perovskite Ceramics. *J Am Chem Soc* **94**, 4091-4107 (2011).
- [4] Zheng Y, Jia X, Li K, Xu J, Bu XH. Energy Conversion in Single-Crystal-to-Single-Crystal Phase Transition Materials. *Adv Energy Mater* **12**, 2100324 (2022).
- [5] Shao Z, Cao X, Luo H, Jin P. Recent progress in the phase-transition mechanism and modulation of vanadium dioxide materials. *NPG Asia Materials* **10**, 581-605 (2018).
- [6] Garvie RC, Hannink R, Pascoe R. Ceramic steel? *Nature* **258**, 703-704 (1975).

3. the solid solution of K in the lattice of Li_2OHCl should be confirmed by some experimental technique such as ICP, SEM-EDS, TEM-EDS, etc.

Reply:

Thank you for your suggestion. Now we have added the SEM-EDS analysis with elemental mapping into this manuscript. As presented in **Fig. 2**, **Fig. 3** and **Table 1**, the K content in $(\text{Li}_2\text{OH})_{1-x}\text{K}_x\text{Cl}$ obtained by SEM-EDS analysis is approximately consistent with the refinement results (based on PND, PXRD, and PDF data).

Fig. 2 The EDS analysis of K in $(\text{Li}_2\text{OH})_{1-x}\text{K}_x\text{Cl}$.

Fig. 3 The morphology of $(\text{Li}_2\text{OH})_{1-x}\text{K}_x\text{Cl}$ ($x = 0.01, 0.05, 0.1$) powder observed by SEM, and EDS mapping images of oxygen, chlorine, and potassium in red, green, and cyan, respectively.

Table 1. K content in antiperovskite $(\text{Li}_2\text{OH})_{1-x}\text{K}_x\text{Cl}$ according to EDS analysis.

x in $(\text{Li}_2\text{OH})_{1-x}\text{K}_x\text{Cl}$	Mol % of K	Error
0.01	2.182	0.497%
0.05	6.617	0.788%
0.1	11.246	1.687%

4. Evaluation of activation energy by BVSE is effective for screening etc. but is difficult to reproduce quantitatively. The energy barriers to migration should be discussed taking into account atomic/electron level interactions such as first-principles calculations.

Reply:

Thank you for your suggestion. To improve the reliability and accuracy of calculations in this manuscript, we performed *ab-initio* molecular dynamics (AIMD) calculation to illustrate the effect of K doping on the ionic migration behaviors, and the difference of migration mechanisms between orthorhombic Li_2OHCl and cubic $(\text{Li}_2\text{OH})_{1-x}\text{K}_x\text{Cl}$. As shown in **Fig. 4** and **Fig. 5**, AIMD calculations embody the advantages of cubic K-doped antiperovskite in three-dimensional migration trajectories. Li atoms trajectories in the orthorhombic Li_2OHCl show a local distribution around the lattice sites, and the mean-squared displacement (MSD) of Li atoms keeps steady with the simulation time. In contrast, Li atoms in cubic K-doped antiperovskite lattice exhibit significant dispersion characteristics and three-dimensional migration trajectories, and the MSD of Li atoms is obviously higher than that of other atoms, which indicates that the ionic conductivity measured in experiments is attributed to Li^+ diffusion. Coincidentally, the nuclear density maps derived from neutron diffraction data using the maximum entropy method (MEM) also confirm the characteristics of the three-dimensional transport path of Li^+ ions in cubic $(\text{Li}_2\text{OH})_{0.99}\text{K}_{0.01}\text{Cl}$ from experimental evidence (**Fig. 6**).

Fig. 4 The AIMD calculations: **a** Trajectories of Li atoms in orthorhombic lattice. **b** Trajectories of Li atoms in cubic K-doped lattice.

Fig. 5 MSD of different atoms in **a** orthorhombic lattice; **b** cubic K-doped lattice.

Fig. 6 2D nuclear density maps of Li_2OHCl and $(\text{Li}_2\text{OH})_{0.99}\text{K}_{0.01}\text{Cl}$ deduced from maximum entropy method analysis. The isosurface level is between -0.02 and $0.04 \text{ fm} \text{ \AA}^{-3}$ in a-c plane, and the arrows indicate the preferable Li^+ ions pathways in both structures. The positive density (O atoms) is displayed as red, and the H atoms with negative density (blue) surround the O atoms. The Li atoms with negative density is also displayed as blue, and marked in the figure by Li1 or Li2.

REVIEWER COMMENTS

Reviewer #1 (Remarks to the Author):

I appreciate the efforts of reviewers. The key finding of this work is the replace of a $[\text{Li}_6\text{OH}]^+$ cluster by a K^+ ion. The authors have presented strong arguments that the conventional substitution of Li^+ by K^+ does not work here. I accept the evidences. But I am not convinced the local coordination of K^+ is stable after it substitutes a $[\text{Li}_6\text{OH}]^+$ cluster.

- (1) Can the authors clearly show the coordination environment of the K^+ ? What are the nearest neighbors? Does the bond valence analysis make sense?
- (2) How about AIMD simulation to further evaluate the stability of the proposed structure?
- (3) Can Cs^+ doping do the same thing?

Reviewer #2 (Remarks to the Author):

In this revised paper, the authors additionally carried out powder neutron diffraction and pair distribution function analysis to investigate the crystal structures and local environment in K-doped Li_2OHCl cubic phases. They also newly discuss the structure stability of K-doped Li_2OHCl by the Goldschmidt tolerance factor parameters from the viewpoint of relative ionic size. In response to the reviewer's point that the BVS analysis was just qualitative, ab initio molecular dynamics (AIMD) calculations were additionally performed, and the diffusion behaviour of each ion was analysed from mean-square displacements. The AIMD simulations show that ion diffusion hardly occurs in the orthorhombic Li_2OHCl phase and Li ions are mobile species in the K-doped Li_2OHCl cubic phase.

The authors properly answer to the reviewers' comments. The content of the paper has improved in quality, as the issues are easier to understand. However, before the publication in the Nature Communications, a few points still need to be revised.

1. According to increment of K dopant concentration, the lattice parameter of K-doped Li_2OHCl become shrink. If K^{+} ions were substituted for Li^{+} ions, the variation of lattice parameters with K-doping is opposite to the fact that the ionic radius of a K^{+} ion is larger than that of a Li^{+} ion. The authors claim that K^{+} dopants replace not Li^{+} ions but $[\text{Li}_6\text{OH}]^{5+}$ octahedral clusters. At first, I would like to point out that the charge state of $[\text{Li}_6\text{OH}]$ cluster is not 1+ but 5+ (6 Li^{+} and 1 $(\text{OH})^{-}$). The cluster $[\text{Li}_6\text{OH}]^{5+}$ and K^{+} has large difference of charge states. Replacement of $[\text{Li}_6\text{OH}]^{5+}$ octahedral cluster composed of 7 ions (8 atoms) by only one K^{+} ion must lead to very large lattice distortion. Such a heavily distorted defect is usually difficult to be formed. The authors carried out AIMD simulations. Although simulated diffusion behavior of Li ion seems to be reliable, results of AIMD calculation rely on its initial composition and configuration. AIMD simulation itself cannot prove that its initial composition and configuration are valid. The validity of the initial structure of AIMD needs to be demonstrated by other means. The possibility of substitutional defects of K^{+} can be judged from the defect formation energy. The calculation methodology of lattice defect formation energy is described by many works such as F. Oba and Y. Kumagai, Appl. Phys. Express, 2018, 11, 060101 or S. Stegmaier et al, Chem. Mater. 2017, 29, 4330. The authors need to show that their proposed defect model is reasonable with quantitative manner.

Following the content of the article, another defect model seems more feasible to me. The authors describe the chemical composition of K-doped Li_2OHCl as $(\text{Li}_2\text{OH})_{1-x}\text{K}_x\text{Cl}$. If one follows this description, one can assume that a $[\text{Li}_2\text{OH}]^{+}$ cluster, 2 Li^{+} ions and 1 $(\text{OH})^{-}$ ion, is replaced by one K^{+} ion.

The author can quantitatively compare from the defect formation energies whether a K^{+} ion is substituted for a $[\text{Li}_6\text{OH}]^{5+}$ cluster or a $[\text{Li}_2\text{OH}]^{+}$ cluster.

2. The authors claim that "Fourthly, both KCl and Li_2OHCl have been experimentally reported to have Pm3m phase, 26, 34, 35 which also implies the formation of solid solution $(\text{Li}_2\text{OH})_{1-x}\text{K}_x\text{Cl}$." This part seems inappropriate. Ref. 34 is the article regarding high pressure crystal structure of potassium chlorate KClO_3 , not KCl. Ref. 35 is the article of high pressure crystal structure of alkali halides. The CsCl-type structure of KCl at 22 kbar is reported. The $(\text{Li}_2\text{OH})_{1-x}\text{K}_x\text{Cl}$ sample is synthesized under an ambient pressure. The crystal structure of KCl at ambient pressure is NaCl-type (Fm-3m). This part

description does not support the validity of K^{+} substitution for the oxygen site in anti-perovskite structure Li_2OHCl .

Reviewers' comments:

Reviewer #1 (Remarks to the Author):

I appreciate the efforts of reviewers. The key finding of this work is the replace of a $[\text{Li}_6\text{OH}]^+$ cluster by a K^+ ion. The authors have presented strong arguments that the conventional substitution of Li^+ by K^+ does not work here. I accept the evidences. But I am not convinced the local coordination of K^+ is stable after it substitutes a $[\text{Li}_6\text{OH}]^+$ cluster.

Reply:

Thank you very much for your comments and recognition of our work, as well as for pointing out the deficiencies to help us further improve this study.

Intuitively, the substitution of $[\text{Li}_6\text{OH}]$ octahedron by K^+ can cause cluster defects in the lattice, leading to the instability of the $(\text{Li}_2\text{OH})_{1-x}\text{K}_x\text{Cl}$ structure. In fact, when x in $(\text{Li}_2\text{OH})_{1-x}\text{K}_x\text{Cl}$ is increased to 0.2, the XRD patterns exhibit a mixture of cubic and orthorhombic phases (as shown in **Figure 1**), indicating that excessive substitution of $[\text{Li}_6\text{OH}]$ octahedral cluster with K^+ may lead to the instability of the cubic $(\text{Li}_2\text{OH})_{1-x}\text{K}_x\text{Cl}$ structure. So, the above experimental phenomenon is also consistent with your viewpoint that the local coordination of K^+ may be unstable after it substitutes the $[\text{Li}_6\text{OH}]$ octahedral cluster (when the K-doping concentration is too high).

Fig.1 The XRD patterns of Li_2OHCl , 10% K-doping sample and 20% K-doping sample respectively.

On the other hand, when the K-doping concentration is within a reasonable range ($0.01 \leq x \leq 0.1$), $(\text{Li}_2\text{OH})_{1-x}\text{K}_x\text{Cl}$ exhibit the single cubic phase structures (**Figure 2**), indicating that $(\text{Li}_2\text{OH})_{1-x}\text{K}_x\text{Cl}$ can tolerate fewer cluster defects and maintain a stable cubic antiperovskite structure. In addition, based on the bond valence model, we calculated the global instability index (GII) for evaluating the stability of cubic antiperovskite $(\text{Li}_2\text{OH})_{1-x}\text{K}_x\text{Cl}$. Usually, the criterion $\text{GII} \leq 0.2$ valence units (v. u.) is used for cubic perovskite structure-in-stability evaluation [1]. Taking $(\text{Li}_2\text{OH})_{0.9}\text{K}_{0.1}\text{Cl}$ as an example, its GII calculation result is 0.112844 v. u., which is within the range of GII values for stable cubic perovskite structures, reflecting that the local coordination of K^+ is stable within the reasonable K-doping range ($0.01 \leq x \leq 0.1$).

Fig.2 The XRD patterns of Li_2OHCl and $(\text{Li}_2\text{OH})_{1-x}\text{K}_x\text{Cl}$ ($0.01 \leq x \leq 0.1$).

References

[1] Feng W, *et al.* Global instability index as a crystallographic stability descriptor of halide and chalcogenide perovskites. *J Energy Chem* **70**, 1-8 (2022).

For responses to your comments/questions, please see below.

1. Can the authors clearly show the coordination environment of the K^+ ? What are the nearest neighbors? Does the bond valence analysis make sense?

Reply:

Thank you for your question.

Based on the results of PDF analysis and Rietveld refinement, it can be approximated that the coordination environment of K^+ in $(Li_2OH)_{1-x}K_xCl$ is similar to that in KCl ($Pm\bar{3}m$). As shown in **Figure 3**, the structural model of $(Li_2OH)_{1-x}K_xCl$ presents that the nearest anion to K^+ is Cl^- , and the partial PDF (**Figure 4**) indicates that K-Cl bond length in $(Li_2OH)_{1-x}K_xCl$ ($0.01 \leq x \leq 0.1$) is from 3.385 to 3.391 Å. According to the database of Materials Project [2], the K-Cl bond length in KCl lattice (space group: $Pm\bar{3}m$, No. mp-23289) is 3.32 Å, which indicates that the K-Cl bond length and coordination environment of K^+ in $(Li_2OH)_{1-x}K_xCl$ structure in this manuscript is reasonable.

Fig.3 The crystal structures of cubic Li_2OHCl , $(Li_2OH)_{1-x}K_xCl$, and KCl ($Pm\bar{3}m$).

Fig.4 a The partial PDF analysis of K-Cl in $(Li_2OH)_{1-x}K_xCl$. **b** The K-Cl bond distance according to partial PDF.

Furthermore, we conducted bond valence sum (BVS) analysis based on the $(Li_2OH)_{0.9}K_{0.1}Cl$ structure according to your suggestion, as shown in **Table 1**. In fact,

the determination of H in crystal structure usually has uncertainty, which may affect the accuracy of BVS analysis for $(\text{Li}_2\text{OH})_{0.9}\text{K}_{0.1}\text{Cl}$. But overall, there is no significant deviation between the values of BVS and the valence state for each atom.

Table 1. Bond valence sum values for antiperovskite $(\text{Li}_2\text{OH})_{0.9}\text{K}_{0.1}\text{Cl}$.

$(\text{Li}_2\text{OH})_{0.9}\text{K}_{0.1}\text{Cl}$, cubic, $\text{Pm}\bar{3}\text{m}$, $a = 3.9023 \text{ \AA}$						
Atoms	Wyckoff	x	y	z	Occ.	BV Sum
Cl	1b	0.5	0.5	0.5	1	-1.044
O	1a	0	0	0	0.888	-2.011
Li	3d	0.5	0	0	0.594	0.902
H1	6e	0.214	0	0	0.049	1.171
H2	12i	0.149	0.149	0	0.056	1.196
K	1a	0	0	0	0.101	0.745

References

[2] Jain A, *et al.* Commentary: The Materials Project: A materials genome approach to accelerating materials innovation. *APL Materials* **1**, 011002 (2013).

2. How about AIMD simulation to further evaluate the stability of the proposed structure?

Reply:

Thank you for your question.

As presented in **Figure 5a**, the energy as a function of time extracted from AIMD simulation shows the stability of the K-doped structure $((\text{Li}_2\text{OH})_{26}\text{KCl}_{27}, 3 \times 3 \times 3$ supercell) at 550 K. Also, we extracted the structure trajectory snapshots of the corresponding phases at 0, 20, and 60 ps, respectively. As shown in **Figure 5b-d**, except for the migrating Li^+ ions, the skeleton atoms of the supercell have been vibrating near the equilibrium position, maintaining the stability of the entire system.

Fig.5 a Evolution of total energy of the proposed K-doped structure. **b-d** The extracted structure during the AIMD simulation at 0, 20, and 60 ps.

3. Can Cs^+ doping do the same thing?

Reply:

Thank you very much for your kind suggestion.

We have attempted the doping of Cs^+ in Li_2OHCl and found that the cubic phase of antiperovskite Li_2OHCl can also be stabilized, achieving a similar effect to K-doping. As shown in **Figure 6a**, the XRD pattern of Li_2OHCl exhibits orthorhombic phase characteristics before Cs-doping. In contrast, the XRD pattern shows obvious cubic phase characteristics after 10% Cs-doping. In addition, it is interesting to note that the XRD peak position of Cs-doped antiperovskite is significantly shifted to the left compared with that of K-doped antiperovskite (**Figure 6b**), which indicates that Cs^+ with a larger ionic radius (1.74 Å) can lead to lattice expansion of cubic antiperovskite.

The above exciting experimental results will prompt us to continue the studies on Cs-doping in antiperovskite in the future.

Fig.6 a The XRD patterns of Li_2OHCl , 10% Cs-doping sample and 10% K-doping sample respectively. **b** The XRD peaks between 40 and 60 degree to illustrate the lattice expansion caused by Cs-doping.

Reviewer #2 (Remarks to the Author):

In this revised paper, the authors additionally carried out powder neutron diffraction and pair distribution function analysis to investigate the crystal structures and local environment in K-doped Li_2OHCl cubic phases. They also newly discuss the structure stability of K-doped Li_2OHCl by the Goldschmidt tolerance factor parameters from the viewpoint of relative ionic size. In response to the reviewer's point that the BVS analysis was just qualitative, ab initio molecular dynamics (AIMD) calculations were additionally performed, and the diffusion behaviour of each ion was analysed from mean-square displacements. The AIMD simulations show that ion diffusion hardly occurs in the orthorhombic Li_2OHCl phase and Li ions are mobile species in the K-doped Li_2OHCl cubic phase.

The authors properly answer to the reviewers' comments. The content of the paper has improved in quality, as the issues are easier to understand. However, before the publication in the Nature Communications, a few points still need to be revised.

Reply:

Thank you very much for your comments and recognition of our work, as well as for pointing out the issues to help us further improve this study. For responses to your comments/questions, please see below.

1. According to increment of K dopant concentration, the lattice parameter of K-doped Li_2OHCl become shrink. If K^+ ions were substituted for Li^+ ions, the variation of lattice parameters with K-doping is opposite to the fact that the ionic radius of a K^+ ion is larger than that of a Li^+ ion. The authors claim that K^+ dopants replace not Li^+ ions but $[\text{Li}_6\text{OH}]^+$ octahedral clusters. At first, I would like to point out that the charge state of $[\text{Li}_6\text{OH}]$ cluster is not 1+ but 5+ (6 Li^+ and 1 $(\text{OH})^-$). The cluster $[\text{Li}_6\text{OH}]^{5+}$ and K^+ has large difference of charge states. Replacement of $[\text{Li}_6\text{OH}]^{5+}$ octahedral cluster composed of 7 ions (8 atoms) by only one K^+ ion must lead to very large lattice distortion. Such a heavily distorted defect is usually difficult to be formed. The authors carried out AIMD simulations. Although simulated diffusion behavior of Li ion seems to be reliable, results of AIMD calculation rely on its initial composition and

configuration. AIMD simulation itself cannot prove that its initial composition and configuration are valid. The validity of the initial structure of AIMD needs to be demonstrated by other means. The possibility of substitutional defects of K^+ can be judged from the defect formation energy. The calculation methodology of lattice defect formation energy is described by many works such as F. Oba and Y. Kumagai, Appl. Phys. Express, 2018, 11, 060101 or S. Stegmaier et al, Chem. Mater. 2017, 29, 4330. The authors need to show that their proposed defect model is reasonable with quantitative manner.

Following the content of the article, another defect model seems more feasible to me. The authors describe the chemical composition of K-doped Li_2OHCl as $(Li_2OH)_{1-x}K_xCl$. If one follows this description, one can assume that a $[Li_2OH]^+$ cluster, 2 Li^+ ions and 1 $(OH)^-$ ion, is replaced by one K^+ ion.

The author can quantitatively compare from the defect formation energies whether a K^+ ion is substituted for a $[Li_6OH]^{5+}$ cluster or a $[Li_2OH]^+$ cluster.

Reply:

Thank you for your kind suggestion. We apologize for the confusing and misleading descriptions of $(Li_2OH)_{1-x}K_xCl$ structure.

We totally agree with you. From the perspective of charge balance and structural stability, the reasonable explanation of K-doping in antiperovskite Li_2OHCl is that K^+ replace $[Li_2OH]^+$ cluster, which leads to the formation of solid solution $(Li_2OH)_{1-x}K_xCl$. In the cubic Li_2OHCl structure, OH^- is bonded to six equivalent Li^+ (with 1/3 vacancies) to form a mixture of vertex and edge-shared $[Li_6OH]$ octahedron (**Figure 1a**). The original meaning of “ K^+ substitute the $[Li_6OH]^+$ octahedral cluster” described in the manuscript refers to the disappearance of the $[Li_6OH]$ octahedral structure formed by the six coordination between OH^- and Li^+ , when K^+ is located at the (0, 0, 0) site. However, from the perspective of single unit cell, there is no octahedral structure in the finite $1 \times 1 \times 1$ lattice, and Li^+ ions (with 1/3 vacancies) only exist on the edges of the cube (**Figure 1b**). Therefore, K-doping in the unit cell of cubic $(Li_2OH)_{1-x}K_xCl$ leads to K^+ substitution of $[Li_2OH]^+$ clusters (**Figure 1c**), which is actually consistent with your viewpoint. Besides, it can be approximated that the coordination environment of

K^+ in the $(Li_2OH)_{1-x}K_xCl$ lattice is similar to that in KCl (Materials Project [1], space group: $Pm\bar{3}m$, No. mp-23289, **Figure 1d**).

Fig.1 a The crystal structure of cubic Li_2OHCl including $[Li_6OH]$ octahedra. **b** The unit cell structure of cubic Li_2OHCl . **c, d** The crystal structure of cubic $(Li_2OH)_{1-x}K_xCl$ and KCl ($Pm\bar{3}m$).

Based on your suggestions, we have realized that the description in the manuscript is misleading, and thank you very much for pointing out this issue. Now we have corrected the inappropriate description of the $(Li_2OH)_{1-x}K_xCl$ structure in the third paragraph in **Structure Determination** section as follows:

“Hence, we infer that K^+ ions substitute O^{2-} ions at the center of the $[Li_6OH]$ octahedra, and Li^+ ions on the edge of the unit cell coordinated with OH^- should be eliminated accordingly considering the charge balance. In other words, as a larger ion, K^+ ions may be located at $(0, 0, 0)$ site in the cubic unit cell and substitute the $[Li_2OH]^+$ cluster.”

Besides, to illustrate the stability of the proposed structure with K^+ substitution for $[Li_2OH]^+$ clusters, we have also calculated the formation energy (E_f) of the defective

structure according to your suggestion. The calculation result of E_f (-0.599 eV) demonstrates that the proposed structure is reasonable. Here, E_f was calculated from the energy difference between cubic Li_2OHCl and K-doped structure (**Equation 1**).

$$E_f = E[(\text{Li}_2\text{OH})_{26}\text{Cl}_{27}\text{K}] + 2 \times E[\text{LiOH}] - E[(\text{Li}_2\text{OH})_{27}\text{Cl}_{27}] - E[\text{KOH}] \quad (1)$$

where $E[(\text{Li}_2\text{OH})_{27}\text{Cl}_{27}]$, $E[(\text{Li}_2\text{OH})_{26}\text{KCl}_{27}]$, $E[\text{LiOH}]$, and $E[\text{KOH}]$ represent the DFT energies of Li_2OHCl supercell ($3 \times 3 \times 3$), K-doped supercell ($3 \times 3 \times 3$), LiOH bulk phase and KOH bulk phase, respectively.

On the other hand, the energy as a function of time extracted from AIMD simulation shows the stability of the K-doped structure at 550 K (**Figure 2a**). In addition, we extracted the structure trajectory snapshots of the corresponding phases at 0, 20, and 60 ps, respectively. As shown in **Figure 2b-d**, except for the migrating Li^+ ions, the skeleton atoms of the supercell have been vibrating near the equilibrium position, maintaining the stability of the entire system.

Fig.2 a Evolution of total energy of the proposed K-doped structure. **b-d** The extracted structure during the AIMD simulation at 0, 20, and 60 ps.

References

[1] Jain A, *et al.* Commentary: The Materials Project: A materials genome approach to accelerating materials innovation. *APL Materials* **1**, 011002 (2013).

2. The authors claim that “Fourthly, both KCl and Li_2OHCl have been experimentally reported to have Pm-3m phase, 26, 34, 35 which also implies the formation of solid solution $(\text{Li}_2\text{OH})_{1-x}\text{K}_x\text{Cl}$.” This part seems inappropriate. Ref. 34 is the article regarding high pressure crystal structure of potassium chlorate KClO_3 , not KCl. Ref. 35 is the article of high pressure crystal structure of alkali halides. The CsCl-type structure of KCl at 22 kbar is reported. The $(\text{Li}_2\text{OH})_{1-x}\text{K}_x\text{Cl}$ sample is synthesized under an ambient pressure. The crystal structure of KCl at ambient pressure is NaCl-type (Fm-3m). This part description does not support the validity of K^+ substitution for the oxygen site in anti-perovskite structure Li_2OHCl .

Reply:

Thank you for your kind suggestion. We agree with your viewpoint that the crystal structure of KCl is NaCl-type (Fm $\bar{3}$ m) at ambient pressure, while KCl with CsCl-type (Pm $\bar{3}$ m) structure can be obtained at 22 kbar.

From the perspective of coordination environment, based on the results of PDF analysis and Rietveld refinement, it can be approximated that the coordination environment of K^+ in $(\text{Li}_2\text{OH})_{1-x}\text{K}_x\text{Cl}$ is similar to that in KCl (Pm $\bar{3}$ m) (as shown in **Figure 1c** and **d**). The structural model of $(\text{Li}_2\text{OH})_{1-x}\text{K}_x\text{Cl}$ presents that the nearest anion to K^+ is Cl^- , and the analysis results of partial PDF (**Figure 3**) indicate that K-Cl bond length in $(\text{Li}_2\text{OH})_{1-x}\text{K}_x\text{Cl}$ ($0.01 \leq x \leq 0.1$) is from 3.385 to 3.391 Å. According to the database of Materials Project [1], the K-Cl bond length in KCl lattice (space group: Pm $\bar{3}$ m, No. mp-23289) is 3.32 Å, which indicates that the K-Cl bond length and coordination environment of K^+ in $(\text{Li}_2\text{OH})_{1-x}\text{K}_x\text{Cl}$ structure in this manuscript is reasonable.

Now we have removed the inappropriate description you mentioned and made adjustments in the third paragraph in **Structure Determination** section as follows:

“Fourthly, the partial PDF data indicate that the nearest anion to K^+ is Cl^- , and it can be approximated that the coordination environment of K^+ in $(Li_2OH)_{1-x}K_xCl$ is similar to that in KCl (space group: $Pm\bar{3}m$, Materials Project, No. mp-23289). Hence, the bonding mode between K^+ and Cl^- also implies the formation of solid solution $(Li_2OH)_{1-x}K_xCl$.”

Fig.3 a The partial PDF analysis of K-Cl in $(Li_2OH)_{1-x}K_xCl$. **b** The K-Cl bond distance according to partial PDF.

REVIEWER COMMENTS

Reviewer #1 (Remarks to the Author):

I appreciate the authors' efforts. However, I am still not convinced of the proposed doping mechanism. In the cubic Li_2OHCl structure, OH^- is bonded to six equivalent Li^+ (with $1/3$ vacancies). Therefore, the composition of each octahedron around OH^- is $[(\text{OH})\text{Li}_4\text{V}_2]^{3+}$, in which V is a vacancy. If a K^+ replace the whole octahedron, the charge is not balanced. The structure of $(\text{Li}_2\text{OH})_{1-x}\text{K}_x\text{Cl}$ in figure 3 of the rebuttal is misleading, because it only shows K^+ replacing 3 sides of an $(\text{OH})^-$ octahedron, not all of the 6 sides.

Reviewer #2 (Remarks to the Author):

The authors' revision comments and the revised article seem to be appropriate. The article is ready for publication, but the authors miss to correct one point for the main text. The authors calculated the formation energy of a K^+ ion substitution for a Li_2OH cluster using the Equation 1 of the comment to the reviewers. This newly added calculation result is not mentioned in the revised main text. It is necessary to include this result to show that the substitutional K^+ defect for the large space of the Li_2OH cluster is a valid model. Please add the formation energy calculation result in the main text before publication,

REVIEWER COMMENTS

Reviewer #1 (Remarks to the Author):

I appreciate the authors' efforts. However, I am still not convinced of the proposed doping mechanism. In the cubic Li_2OHCl structure, OH^- is bonded to six equivalent Li^+ (with 1/3 vacancies). Therefore, the composition of each octahedron around OH^- is $[(\text{OH})\text{Li}_4\text{V}_2]^{3+}$, in which V is a vacancy. If a K^+ replace the whole octahedron, the charge is not balanced. The structure of $(\text{Li}_2\text{OH})_{1-x}\text{K}_x\text{Cl}$ in figure 3 of the rebuttal is misleading, because it only shows K^+ replacing 3 sides of an $(\text{OH})^-$ octahedron, not all of the 6 sides.

Reply:

Thank you very much for your comments and pointing out this issue to help us further improve this study. We apologize for the unclear description of the doping mechanism in the manuscript.

We completely agree with you. If a K^+ replaces the whole octahedron, the charge is not balanced. The reasonable explanation of K-doping in Li_2OHCl is that K^+ is located at the (0, 0, 0) site, and replaces the OH^- and three Li^+ (with 1/3 vacancies) surrounding OH^- , that is, K^+ replaces $[\text{Li}_2\text{OH}]^+$ cluster (or $[\text{Li}_2\text{V}_1\text{OH}]^+$, in which V is a vacancy) rather than $[\text{Li}_4\text{V}_2\text{OH}]^{3+}$ octahedron. In this case, K-doping leads to the formation of solid solution $(\text{Li}_2\text{OH})_{1-x}\text{K}_x\text{Cl}$, and the charge is balanced.

The original meaning of “ K^+ substitute the octahedron cluster” (we realize that this description is actually incorrect) is that K^+ is located at the (0, 0, 0) site and replaces the $[\text{Li}_2\text{V}_1\text{OH}]^+$ cluster in $[\text{Li}_4\text{V}_2\text{OH}]$ octahedron, leading to that the original six coordination structure between OH^- and Li^+ (with 1/3 vacancies) in octahedron no longer exists. In other words, the substitution of K^+ for $[\text{Li}_2\text{V}_1\text{OH}]^+$ cluster leads to the disappearance of $[\text{Li}_4\text{V}_2\text{OH}]$ (or $[\text{Li}_6\text{OH}]$ with one-third of Li vacancies) octahedron in the local structure, but this is not equivalent to K^+ replacing the whole octahedron. In this case, the two unsubstituted Li^+ (including one vacancy) of $[\text{Li}_4\text{V}_2\text{OH}]$ octahedron remain in the lattice and still coordinate with OH^- in other adjacent octahedra. In addition, the partial PDF analysis indicate that the nearest ion to K^+ is Cl^- (as shown in Fig. S4 of Supplementary Information and the previous reply), so the two unsubstituted

Li^+ are not at the sites near K^+ , but at the sites far from K^+ and coordinate with OH^- in the adjacent octahedron (each octahedron originally has one-third of Li vacancies). Therefore, the local coordination of K^+ is stable and $(\text{Li}_2\text{OH})_{1-x}\text{K}_x\text{Cl}$ with K^+ substituted $[\text{Li}_2\text{OH}]^+$ cluster is charge balanced overall.

It is worth pointing out, in the AIMD calculations of this study, the $3 \times 3 \times 3$ supercell structure model $(\text{Li}_2\text{OH})_{26}\text{KCl}_{27}$ was created based on the doping mechanism of K^+ replacing $[\text{Li}_2\text{OH}]^+$ cluster rather than the whole octahedron, and its stability has been evaluated according to the evolution of total energy in the previous reply. Therefore, both from the perspective of structural stability and charge balance, as well as the validation experiments of K-doping in Li_2OHBr and Cs-doping in Li_2OHCl , all of which can demonstrate the possibility and rationality of the proposed doping mechanism of K^+ replacing $[\text{Li}_2\text{OH}]^+$ cluster in $(\text{Li}_2\text{OH})_{1-x}\text{K}_x\text{Cl}$.

Based on your comments, we have realized that the description of the K-doping mechanism in this manuscript is misleading and ambiguous, and thank you very much for pointing it out. At present, we have clarified the mechanism of K-doping, and there is no inappropriate description of K^+ replacing octahedron in the manuscript. In addition, we have corrected the description of $(\text{Li}_2\text{OH})_{1-x}\text{K}_x\text{Cl}$ structure in the third paragraph in **Structure Determination** section as follows:

“Hence, we infer that K^+ ions may be located at (0, 0, 0) site and replace the $[\text{Li}_2\text{OH}]^+$ cluster in $[\text{Li}_6\text{OH}]$ octahedron (with one-third of Li vacancies), which leads to the disappearance of the original six coordination structure between OH^- and Li^+ (**Fig. 3d** and **e**). It should be noted that the two unsubstituted Li^+ of $[\text{Li}_6\text{OH}]$ octahedron remain in the lattice and still coordinate with OH^- in other adjacent octahedra, and $(\text{Li}_2\text{OH})_{1-x}\text{K}_x\text{Cl}$ with K^+ substituted $[\text{Li}_2\text{OH}]^+$ cluster is charge balanced overall.”

Reviewer #2 (Remarks to the Author):

The authors' revision comments and the revised article seem to be appropriate. The article is ready for publication, but the authors miss to correct one point for the main text. The authors calculated the formation energy of a K^+ ion substitution for a Li_2OH cluster using the Equation 1 of the comment to the reviewers. This newly added calculation result is not mentioned in the revised main text. It is necessary to include this result to show that the substitutional K^+ defect for the large space of the Li_2OH cluster is a valid model. Please add the formation energy calculation result in the main text before publication.

Reply:

Thank you for your kind suggestion. Now the calculation result of formation energy to demonstrate the validity of $(Li_2OH)_{1-x}K_xCl$ structure model has been added in **Structure Determination** section as follows:

“Also, the formation energy (E_f) of $(Li_2OH)_{26}KCl_{27}$ supercell ($3 \times 3 \times 3$) is calculated to be -0.599 eV, indicating that the proposed structural model of K^+ replacing $[Li_2OH]^+$ cluster in $(Li_2OH)_{1-x}K_xCl$ is reasonable.”

Besides, the computational method for the formation energy calculation has been added in **Methods** section as follows:

“The formation energy (E_f) is calculated from the energy difference between cubic Li_2OHCl and K-doped structure (**Equation 2**) to demonstrate the validity of $(Li_2OH)_{1-x}K_xCl$ structure model.

$$E_f = E[(Li_2OH)_{26}Cl_{27}K] + 2 \times E[LiOH] - E[(Li_2OH)_{27}Cl_{27}] - E[KOH] \quad (2)$$

where $E[(Li_2OH)_{27}Cl_{27}]$, $E[(Li_2OH)_{26}KCl_{27}]$, $E[LiOH]$, and $E[KOH]$ represent the DFT energies of Li_2OHCl supercell ($3 \times 3 \times 3$), K-doped supercell ($3 \times 3 \times 3$), $LiOH$ bulk phase and KOH bulk phase, respectively.”